# Orientation control strategies and adaptation to a visuomotor perturbation in rotational hand movements

Or Zruya[1]*, Ilana Nisky[1,2]

**1** Department of Biomedical Engineering, Ben Gurion University of the Negev, Beer Sheva, Israel,
**2** Department of Engineering, University of Cambridge, Cambridge, United Kingdom

\* orzr@post.bgu.ac.il

**Data Availability Statement:** All human data and statistical analysis code are available at https://github.com/Bio-Medical-Robotics-BGU/Orientation-control-strategies.

## Abstract

Computational approaches to biological motor control are used to discover the building blocks of human motor behaviour. Models explaining features of human hand movements have been studied thoroughly, yet only a few studies attempted to explain the control of the orientation of the hand; instead, they mainly focus on the control of hand translation, predominantly in a single plane. In this study, we present a new methodology to study the way humans control the orientation of their hands in three dimensions and demonstrate it in two sequential experiments. We developed a quaternion-based score that quantifies the geodicity of rotational hand movements and evaluated it experimentally. In the first experiment, participants performed a simple orientation-matching task with a robotic manipulator. We found that rotations are generally performed by following a geodesic in the quaternion hypersphere, which suggests that, similarly to translation, the orientation of the hand is centrally controlled, possibly by optimizing geometrical properties of the hand's rotation. This result established a baseline for the study of human response to perturbed visual feedback of the orientation of the hand. In the second experiment, we developed a novel visuomotor rotation task in which the rotation is applied on the hand's rotation, and studied the adaptation of participants to this rotation, and the transfer of the adaptation to a different initial orientation. We observed partial adaptation to the rotation. The patterns of the transfer of the adaptation to a different initial orientation were consistent with the representation of the orientation in extrinsic coordinates. The methodology that we developed allows for studying the control of a rigid body without reducing the dimensionality of the task. The results of the two experiments open questions for future studies regarding the mechanisms underlying the central control of hand orientation. These results can be of benefit for many applications that involve fine manipulation of rigid bodies, such as teleoperation and neurorehabilitation.

## Author summary

Daily motor actions, as simple as pouring a glass of wine or as complicated as playing a violin, require coordinated activation of multiple muscles that synchronize to produce a

**Funding:** OZ and IN were supported by the Israel Science Foundation (grant number 327/20, http://www.isf.org.il). The funders had no role in study design, data collection and analysis, decision to publish, or preparation of the manuscript.

**Competing interests:** The authors have declared that no competing interests exist.

precise motion of the hand. Controlled by sensorimotor areas in the central nervous system, our muscles can translate and rotate our hand from one posture to another. Our study focused on the very basis of the control of orientation: using a novel methodology, we attempted to reveal which variables are centrally controlled when we rotate our hand in three dimensions. The discovery that simple rotations are generally performed along a geometrically optimal path established a baseline for studying the response to rotation-based perturbations. By visually remapping the orientation of the hand, we found that humans perceive their hand's orientation in visual, rather than joint-based coordinates. These findings have implications for the design of human-centered control systems for teleoperation, where visual distortions may occur, and for the design of rehabilitation devices for people with motor impairments.

## Introduction

Precise control of the position and orientation of the hand is key to accurately performing daily object manipulation tasks. To discover the building blocks of human motor control, experimental and computational approaches may be used. These building blocks are often considered to be motor invariants—robust patterns that can be quantified in human movements. They are typically observed across repetitions of the same movements within and between participants, and are considered to be the result of active control by the nervous system [1]. One such invariant is the straight path and bell-shaped velocity trajectory that characterize fast point-to-point movement; these may result from the optimization of movement smoothness [1–4]. This invariant may also suggest that point-to-point hand movements are primarily under kinematic control in extrinsic coordinates that minimizes reaching errors, consistent with models of deterministic optimal control [1]. This hypothesis is reinforced by the robustness of the invariant to visual [5, 6] and dynamical [7, 8] distortions, and the adaptation in the presence of these distortions is explained with state-space models [9–12]. Alternative non-kinematic hypotheses were also suggested to predict the invariant point-to-point arm movements. Such are models of stochastic optimal feedback control [13–15], which minimize efforts in the presence of signal dependent noise, and the minimum torque change model [16], which accounts for the dynamics of the arm. A recent study presented a model that explains both smoothness and duration of reaching movements in terms of economy by considering the expenditure of metabolic energy [17]. Other studies claimed that control strategies also depend on the cognitive perception of the task. For example, straight paths in joint space were observed when a two-link arm was controlled, rather than just the endpoint position [18].

Controlling motion involves the mapping of sensory input to motor output. It is generally held that this mapping is closely tied to the acquisition of an internal model of the motor apparatus [7, 19, 20]. The internal model may be composed of a forward model that predicts sensory consequences of a given motor command [21, 22], and an inverse model that specifies the motor commands required to produce a desired sensory output. The internal model may be used to compensate for estimation error [13], and to stabilize the control system [23]. Important supporting evidence for the existence of internal models is motor adaptation – the recovery of performance in response to a changed environment [7, 24]. The nature of the internal models is often investigated by studying motor adaptation in the face of perturbations and their aftereffects, generalization, and transfer upon removal of the perturbation. For example, adaptation to visuomotor rotation—a remapping between hand movement and visual display

—was shown to be learned in extrinsic, rather than intrinsic, hand coordinates [24], although mixed coordinates have also been proposed [25, 26].

In contrast to the control of translational movements allowing for the transfer of objects from one position in space to another, the control of rotational movements allowing for a change of the orientation of objects from one orientation to another has not received much attention. Studies of reach to grasp movements reported features similar to two-dimensional translations, such as typical bell-shaped angular velocity profiles [27], and showed evidence of grasp anticipation and orientation online corrections [28]. Moreover, hand orientation for grasping was found to be coupled with reaching direction [29, 30] and target orientation [31], but also related to the posture of the arm, suggesting that both extrinsic and intrinsic variables are controlled [32, 33]. The hypothesis that position and orientation are controlled simultaneously was supported by a gradient-based model that predicted the coarticulation of hand translation and rotation along a path generated during reach to grasp [34]. Nevertheless, evidence for separate control of position and orientation in reach to grasp were also observed [31, 35]. For example, parallel control channels for position and orientation were evidenced by the lack of variation in variable errors between orientation-matching with and without hand translation [32], which supports the three-component hypothesis—prehension movements are controlled through channels for the translation, the manipulation, and the orientation of the hand [36].

Two-dimensional wrist rotations were the main focus in studies of pointing movements that mapped the orientation of a constrained wrist to a position of a point cursor on a screen [37–39]. One such study found that the projected paths of the wrist rotations are more variable and curved than planar reaching movements, which may suggest that wrist rotations are not under kinematic control. Yet, the difference in the curvature of the projected wrist rotation of inbound and outbound pointing suggested that these results are an outcome of imperfect peripheral execution of the motor command [38]. Other studies found robust motor invariants in kinematically redundant moderate pointing movements that involve both the wrist and the forearm, such as the fact that the cursor's path follows Donders' law for dimensionality reduction [40–42]. These stereotypical patterns were later attributed to strategies that use wrist flexion-extension and wrist radial-ulnar deviations, but not forearm pronation-supination [43]. A few studies also attempted to investigate adaptation to perturbations during wrist pointing. For example, a study of torque field adaptation found evidence for central control of orientation, as the projected path's curvature increased when the torque field was suddenly reversed [37]. Another study that employed visuomotor rotation on the cursor's path showed that a phase shift between the intrinsic (body-fixed) and extrinsic (space-fixed) frames worsened adaptation [39]. This adaptation was later shown to be compensated and narrowly generalized to different directions [44], similarly to visuomotor rotation in studies of point-to-point movements [24].

Common to these wrist pointing studies is the mapping from hand orientation to cursor position visualized on a two-dimensional screen. However, these studies do not tell us how humans control three-dimensional rotations of the hand while manipulating a three-dimensional object, and how they adapt to visuomotor perturbations in such rotations. A theoretical study exploited dual quaternions—a mathematical object used to represent movement in six degrees of freedom—to describe three-dimensional kinematics, and presented its use in models predicting the motion of the hand [45]. Another study of rotation gain adaptation provided three-dimensional visual feedback by aligning the orientation of a manipulated virtual object to that of the hand, and revealed the construction of a transferable internal model [46]. However, except for these two studies, the control the orientation of a rigid body which is held by the hand remains largely unknown.

In this study, we aimed to bridge this gap, and investigated hand orientation control and adaptation without reducing the degrees of freedom of the object's orientation. We proposed a new methodology to study the control of the orientation of a rigid body. To provide a realistic three-dimensional visual feedback, we visualized the orientation of the controlled object using a three-dimensional cube. Unlike in previous studies that analyzed the orientation trajectory by projecting the orientation of the hand on a two-dimensional plane [37–39], we represent the orientation trajectory using quaternions—one of the common representations of orientation. We demonstrated the new methodology in two sequential experiments. First, we approached the fundamental question of whether simple hand rotations are under kinematic control of the sensorimotor system. We therefore developed a score to quantify the geodicity of rotational hand movements in quaternion space and evaluated it in an orientation-matching task. If rotations are indeed centrally controlled, we expected to observe evidence of path planning optimization. Indeed, we found a tendency to follow geodesics, which established a baseline of response to perturbed hand orientation, and inspired our second question of interest— in which coordinate system is the orientation of a rigid body represented? We developed a novel visuomotor rotation task in which the rotation of a simulated controlled rigid body is perturbed; By remapping the orientation of the hand and applying adaptation and generalization paradigms, we tested whether humans use extrinsic or intrinsic coordinates. Under each hypothesis, we expected to observe transfer of adaptation to a different initial hand orientation when the perturbation was learned and transferred to similar targets in extrinsic or intrinsic coordinates. Our results are consistent with the hypothesis that humans use extrinsic coordinates.

## Methods

### Ethics statement

All participants in this study signed an informed consent form approved by the Human Participants Research Committee of Ben-Gurion University of the Negev, Beer-Sheva, Israel.

### Notation

Throughout this paper, scalar values are denoted by small italic letters (e.g., $x$), vector values are denoted by small bold letters (e.g., $\mathbf{x}$), matrices are denoted by large bold letters (e.g, $\mathbf{X}$), and geometrical spaces are denoted by blackboard letters (e.g., $\mathbb{X}$).

### Mathematical formulation

The manipulation of a rigid body in three-dimensional space involves movement with six degrees of freedom. Three degrees of freedom are associated with translation—the movement of the origin of a reference frame that is attached to the body, and the other three are associated with rotation of the reference frame about an axis. Pose-to-pose movements require the hand to accurately translate and rotate a rigid body to a target pose. Such six-degree-of-freedom movement is composed of point-to-point movement (i.e., the translation of a rigid body between two points in space) and of an orientation-matching movement (i.e., the rotation of a rigid body between two orientations in space). Our study investigates the rotational component of the movement. We considered rotations represented by unit quaternions, denoted by $\mathbf{q} \in \mathbb{H}_1$. $\mathbb{H}_1$ is the space of unit quaternions and is also known as the three-sphere—a four-dimensional unit size sphere. Quaternions are hyper-complex numbers comprised of a real part $s \in \mathbb{R}$ and an imaginary part $\mathbf{v} = [x, y, z] \in \mathbb{R}^3$, such that $\mathbf{q} = s + \hat{\mathbf{i}}x + \hat{\mathbf{j}}y + \hat{\mathbf{k}}z \in \mathbb{H}_1$, $\hat{\mathbf{i}}^2 = \hat{\mathbf{j}}^2 = \hat{\mathbf{k}}^2 = \hat{\mathbf{i}}\hat{\mathbf{j}}\hat{\mathbf{k}} = -1$, $\hat{\mathbf{i}}\hat{\mathbf{j}} = \hat{\mathbf{k}}$ and $\hat{\mathbf{j}}\hat{\mathbf{i}} = -\hat{\mathbf{k}}$. According to Euler's theorem, for any $\mathbf{q} \in \mathbb{H}_1$,

there exists an angle $\theta \in (-\pi, \pi]$ and an axis $\hat{\mathbf{n}} \in \mathbb{R}^3$, such that:

$$\mathbf{q} = \cos \theta/2 + \hat{\mathbf{n}} \sin \theta/2, \tag{1}$$

where $\mathbf{q}$ is the quaternion that takes a point $\mathbf{p} \in \mathbb{R}^3$ and rotates it $\theta$ around $\hat{\mathbf{n}}$ to get $\mathbf{p}'$:

$$\mathbf{p}' = \mathbf{q}\mathbf{p}\mathbf{q}^{-1}. \tag{2}$$

Let $\{\mathbf{q}\}_{i=1}^N$ be a discrete time $N$-sample orientation trajectory (e.g., of the handle of the robotic manipulator—see the Experimental setup and software section for details), such that $\mathbf{q}_i = \cos \theta_i/2 + \hat{\mathbf{n}}_i \sin \theta_i/2$ is the quaternion that rotates all the vectors represented in an identity frame (i.e., with orientation $\mathbf{q}_I = 1$) by an angle $\theta_i$ around an axis $\hat{\mathbf{n}}_i$. We denote the discrete time trajectory of the transition quaternion as $\{\mathbf{q}^\delta\}_{i=1}^{N-1}$, such that $\mathbf{q}_i^\delta = \cos \theta_i^\delta/2 + \hat{\mathbf{n}}_i^\delta \sin \theta_i^\delta/2$ is the quaternion that rotates $\mathbf{q}_i$ to $\mathbf{q}_{i+1}$ by an instantaneous angle $\theta_i^\delta$ around an instantaneous axis $\hat{\mathbf{n}}_i^\delta$. The instantaneous rotation axes intersect in the origin of the coordinate system (the center of mass of the controlled body), and can be represented in either extrinsic or intrinsic coordinates:

$$\text{Extrinsic representation}: \quad \mathbf{q}_{i+1} = \mathbf{q}_i^\delta \mathbf{q}_i, \tag{3}$$

$$\text{Intrinsic representation}: \quad \mathbf{q}_{i+1} = \mathbf{q}_i \mathbf{q}_i^\delta. \tag{4}$$

The product is generally different since quaternion multiplication is not commutative.

Two-dimensional translations—the well-studied reaching movements—are performed by following a straight path in extrinsic coordinates. In differential geometry, the straight path is considered to be a geodesic in $\mathbb{R}^2$—the shortest path that connects two points in the plane. Since all unit quaternions lie on the three-sphere, a geodesic in $\mathbb{H}_1$ is the shortest great arc connecting two quaternions. A geodesic in $\mathbb{H}_1$ between $\mathbf{q}_1$ and $\mathbf{q}_N$ is comprised of all quaternions $\mathbf{q}_g \in \mathbb{H}_1$ that are parametrized by $h \in [0, 1]$ and obey [47]:

$$\mathbf{q}_g(\mathbf{q}_1, \mathbf{q}_N, h) = \mathbf{q}_1(\mathbf{q}_1^* \mathbf{q}_N)^h, \tag{5}$$

where $\mathbf{q}_1^* = \mathbf{q}_1^{-1}$ is the conjugate of $\mathbf{q}_1$. A geodesic between $\mathbf{q}_1 = \mathbf{q}_I$ and $\mathbf{q}_N = \cos \theta_N/2 + \hat{\mathbf{n}}_N \sin \theta_N/2$ results in the following path:

$$\begin{aligned} \mathbf{q}_g(\mathbf{q}_I, \mathbf{q}_N, h) &= \mathbf{q}_I(\mathbf{q}_I^* \mathbf{q}_N)^h \\ &= \mathbf{q}_N^h \\ &= \cos h\theta_N/2 + \hat{\mathbf{n}}_N \sin h\theta_N/2. \end{aligned} \tag{6}$$

Eq 6 suggests that a geodesic in $\mathbb{H}_1$ is achieved by scaling $\theta_N$ according to $h$, and that the axis from Eq 1 is constant and is equal to $\hat{\mathbf{n}}_N$. However, this does not generalize to initial orientations that differ from $\mathbf{q}_I$. For that reason, when we analyzed quaternion curves, we first rotated all quaternions by the inverse of the initial orientation. This is justified since the rotation operation does not alter the geodicity of a quaternion curve, such that for any $\mathbf{q}_1 \in \mathbb{H}_1$:

$$\mathbf{q}_g(\mathbf{q}_1, \mathbf{q}_N, h) = \mathbf{q}_1 \mathbf{q}_g(\mathbf{q}_I, \mathbf{q}_1^{-1}\mathbf{q}_N, h). \tag{7}$$

By inserting Eq 5 into the right side of Eq 7 we get:

$$
\begin{aligned}
\mathbf{q}_1 \mathbf{q}_g(\mathbf{q}_I, \mathbf{q}_1^{-1}\mathbf{q}_N, h) &= \mathbf{q}_1 \mathbf{q}_I (\mathbf{q}_I^* \mathbf{q}_1^{-1} \mathbf{q}_N)^h \\
&= \mathbf{q}_1 (\mathbf{q}_1^{-1} \mathbf{q}_N)^h \\
&= \mathbf{q}_1 (\mathbf{q}_1^* \mathbf{q}_N)^h \\
&= \mathbf{q}_g(\mathbf{q}_1, \mathbf{q}_N, h).
\end{aligned}
\tag{8}
$$

We can also acquire the transition quaternion trajectory ($\{\mathbf{q}^\delta\}_{i=1}^{N-1}$) for a geodesic by inserting Eq 6 into Eqs 3 or 4:

$$
\begin{aligned}
\mathbf{q}_i^\delta &= \mathbf{q}_N^{h_{i+1}} \mathbf{q}_N^{-h_i} \\
&= \mathbf{q}_N^{\Delta h_i} \\
&= \cos \Delta h_i \theta_N / 2 + \hat{\mathbf{n}}_N \sin \Delta h_i \theta_N / 2,
\end{aligned}
\tag{9}
$$

where $\Delta h_i = h_{i+1} - h_i$. Eq 9 suggests that the instantaneous rotation angle $\theta_i^\delta$ is scaled according to $\Delta h_i$, and that the instantaneous rotation axis is constant and equal to $\hat{\mathbf{n}}_N$.

In experiment 1, we developed a quaternion-based score that quantifies the geodicity of rotational hand movements and evaluated it in an orientation-matching experiment. First, we defined the angular distance between two quaternions $\mathbf{q}_i$ and $\mathbf{q}_{i+1}$ using the angle of the transition quaternion:

$$
\text{dist}(\mathbf{q}_i, \mathbf{q}_{i+1}) = 2 \arccos |\, \Re(\mathbf{q}_{i+1}\mathbf{q}_i^{-1}) \,|,
\tag{10}
$$

where the distance is bounded to $[0, \pi)$. Additionally, $\text{dist}(\mathbf{q}_i, -\mathbf{q}_i) = 0$, as $\mathbf{q}_i$ and $-\mathbf{q}_i$ are equal by definition. Then, we defined the Quaternion Geodicity Score (QGS) of a curve $\{\mathbf{q}\}_{i=1}^{N}$ as follows:

$$
\text{QGS} = \frac{\sum_{i=1}^{N-1} \text{dist}(\mathbf{q}_i, \mathbf{q}_{i+1})}{\text{dist}(\mathbf{q}_1, \mathbf{q}_N)} \in [1, \infty).
\tag{11}
$$

A geodesic in $\mathbb{H}_1$ yields QGS = 1. This is proved by inserting Eqs 9 and 10 into Eq 11 and using the sum of a telescopic sequence:

$$
\begin{aligned}
\text{QGS} &= \frac{\sum_{i=1}^{N-1} 2 \arccos |\, \Re(\mathbf{q}_{i+1}\mathbf{q}_i^{-1}) \,|}{2 \arccos |\, \Re(\mathbf{q}_N\mathbf{q}_I^{-1}) \,|} \\
&= \frac{\sum_{i=1}^{N-1} \Delta h_i \theta_N}{\theta_N} \\
&= h_N - h_1 = 1.
\end{aligned}
\tag{12}
$$

Eq 11 suggests that the QGS of a rotational movement is the ratio between the angular displacement accumulated during the movement and the angular displacement achieved by a geodesic. A non-geodesic receives QGS > 1, as it requires that the orientation of the rigid body would deviate from the great-arc that connects the initial and final quaternions, resulting in a longer path to be traveled in $\mathbb{H}_1$.

## Experimental setup and software

The experimental setup consisted of a developed simulation in a virtual environment (VE) using CHAI3D 3.2 API, written in C++ (Visual Studio 2013, Microsoft) on a HP Z440 PC running Windows 10 OS (Microsoft). Participants viewed the VE through a three-dimensional viewer HMZ-T3W (Sony) and interacted with it using the handle of a SIGMA 7 (Force Dimension) robotic manipulator. The three-dimensional viewer was mounted on a metal frame and directed at 45˚ towards the robotic handle (Fig 1a). We implemented a haptic thread rendered at 4 [kHz] and a visual thread rendered at 60 [Hz], simultaneously. To enable a three-dimensional view of the VE, we presented each eye with a visual resolution of 1080p.

The participants sat in front of the robotic manipulator, looked into the three-dimensional viewer and held the handle of the robotic manipulator with their right hand. When the participant's hand was oriented as $q_I$, $x^+$ was directed from the handle towards the participant's body, $y^+$ was directed from the handle to the right hand side of the participant and $z^+$ was directed from the handle upwards. To secure the participant's hand to the handle, we fitted two velcro loops around their thumb and index fingers (Fig 1a). Prior to the beginning of the experiments, we introduced the VE to the participants and briefed them regarding how to hold the handle.

## Procedure and protocol

**Experiment 1.** The VE consisted of a virtual cursor cube (Fig 1b) positioned at the center of the VE. Participants controlled the orientation of the cursor by rotating the handle of the robotic manipulator. To provide veridical visual feedback of the orientation of the handle, we did not apply any transformation between the orientation of the handle and the cursor. Participants only controlled the orientation of the cursor and not its position, which remained constant throughout the experiment. To break the cube's symmetry, we colored each of the side faces differently. We directed the virtual camera at 45˚ towards the cube to match the direction of the three-dimensional viewer with respect to the handle. We did not constrain the movement of the arm, such that participants were able to utilize the shoulder, elbow and wrist joints. Additionally, we applied gravity compensation to support the hand and the robotic handle

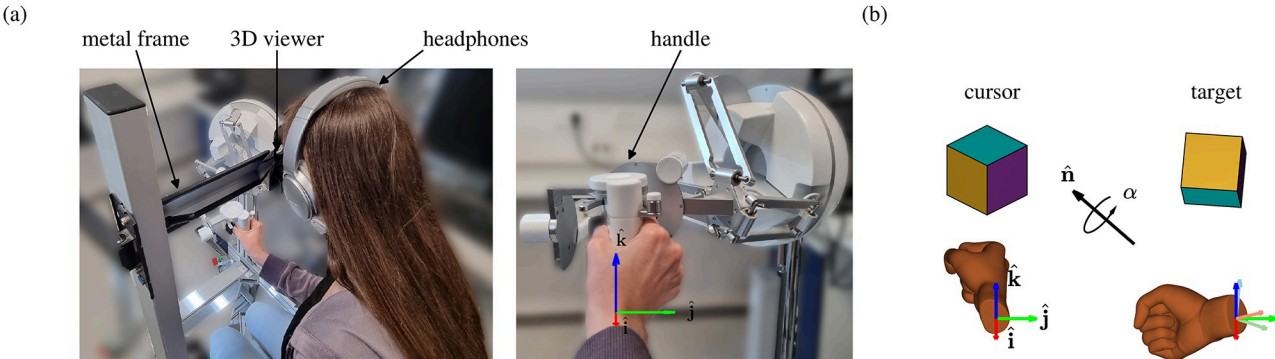

**Fig 1. Experimental setup and orientation-matching task.** (a) Participants were seated in front of a robotic manipulator and interacted with it using their right hand. We projected the VE onto the three-dimensional viewer that was mounted on a metal frame and directed at 45˚ towards the robotic manipulator. To avoid external interruptions, participants wore noise cancelling headphones. (b) Participants controlled the orientation of a cursor cube that was positioned at the center of the VE by rotating their right hand. They were instructed to match the orientation of the cursor cube to that of a target cube, which was rotated $\alpha$ around $\hat{n}$ with respect to the initial cursor orientation. Examples of initial cursor orientation (left) and target orientation (right) are illustrated, along with the orientation of the participant's right hand. The dark red, green and blue arrows depict an extrinsic reference frame, and the light arrows depict an intrinsic reference frame (the frames are aligned when the hand is oriented $q_I$, as in the initial orientation shown here).

using a translational operational space PD controller with a proportional gain $K_p = 100$ [N/m] and a linear damping gain $K_D = 30$ [N · s/m]. Prior to each trial, the same controller was used to translate the hand of the participant to the initial orientation.

Fourteen right handed volunteers participated in experiment 1 (11 males and 3 females, all in the age range of 24–34). The participants were naive with regard to the purpose of the experiments and were reimbursed for their participation.

The experiment consisted of 408 trials, in which we instructed the participants to match the orientation of the cursor cube with that of a second identical cube, referred to as the target, by rotating the handle of the robotic manipulator using their right hand. To minimize external influence on the formed orientation trajectories, we did not instruct the participants to follow any strategy nor did we limit the trial's duration. At the beginning of each trial, a fixed target appeared in one of four random positions in the y-z plane of the VE—above, below, to the right or to the left of the cursor. To avoid participants getting used to a certain target position, we pseudo-randomized their order beforehand, such that targets appeared at each position in an equal number of trials. To present the target, we rotated the initial orientation of the cursor by an angle $\alpha$ around an axis $\hat{\mathbf{n}}$ (Fig 1b). In each trial, we sampled the rotation angle pseudo-randomly from a uniform distribution, i.e., $\alpha \sim U$ [40˚, 60˚]. Participants were cued to initiate their movement when the target appeared. We ended the trial when the angular distance between the orientations of the cursor and of the target, as defined in Eq 10, decreased below 10˚ for a period of 100 [ms], or when it decreased below 15˚ for a period of 400 [ms]. We added the latter, more lenient, condition to avoid frustration among participants. As each trial ended, we removed the target and turned the cursor red, while the robotic manipulator autonomously translated and rotated to the initial position and orientation for the subsequent trial. To reorient the handle, we applied three-degree-of-freedom torques using a rotational operational space PD controller with a proportional gain $K_p = 0.5$ [N·m/rad] and an angular damping gain $K_D = 0.07$ [N·m/s·rad]. To provide smooth autonomous rotation, we provided the handle with a minimum-jerk input torque signal. We initiated the next trial when the angular distance between the orientation of the handle and the initial orientation of the next trial decreased below 5˚ for a period of 100 [ms]. Then, we colored the faces of the cube as before and presented the participants with a new target.

Each orientation-matching trial was defined by an initial orientation and a target orientation. To test the effect of the initial orientation of the hand on the geodicity of the path, we chose the initial orientation of the hand to be one of two orientations in half of the trials ($\mathbf{o}_1$ and $\mathbf{o}_2$, Table 1). Similarly, to test the effect of the target orientation, we chose it to be one of two orientations in the other half of the trials ($\mathbf{t}_1$ and $\mathbf{t}_2$, Table 1). When we chose the initial orientation, the target orientation was determined by rotating the initial orientation around an axis (Table 1) by an angle $\alpha$. When we chose the target orientation, the initial orientation was determined by inversely rotating the target orientation around an axis by an angle $\alpha$. In

**Table 1. Initial cursor and target orientations and rotation axes.**

| Initial orientation | Target orientation |
|---|---|
| $\mathbf{o}_1 = 0.92 + 0.33\hat{\mathbf{i}} - 0.06\hat{\mathbf{j}} + 0.16\hat{\mathbf{k}}$ | $\mathbf{t}_1 = 0.84 - 0.13\hat{\mathbf{i}} - 0.22\hat{\mathbf{j}} + 0.48\hat{\mathbf{k}}$ |
| $\mathbf{o}_2 = 0.92 + 0.16\hat{\mathbf{i}} - 0.06\hat{\mathbf{j}} + 0.33\hat{\mathbf{k}}$ | $\mathbf{t}_2 = 0.9 - 0.04\hat{\mathbf{i}} - 0.08\hat{\mathbf{j}} + 0.42\hat{\mathbf{k}}$ |
| **Aligned axes** | **Misaligned axes** |
| $\hat{\mathbf{i}}$ | $\hat{\mathbf{i}}/\sqrt{2} + \hat{\mathbf{k}}/\sqrt{2}$ |
| $-\hat{\mathbf{j}}$ | $-\hat{\mathbf{j}}/\sqrt{2} + \hat{\mathbf{k}}/\sqrt{2}$ |
| $\hat{\mathbf{k}}$ | $\hat{\mathbf{i}}/\sqrt{2} - \hat{\mathbf{j}}/\sqrt{2}$ |

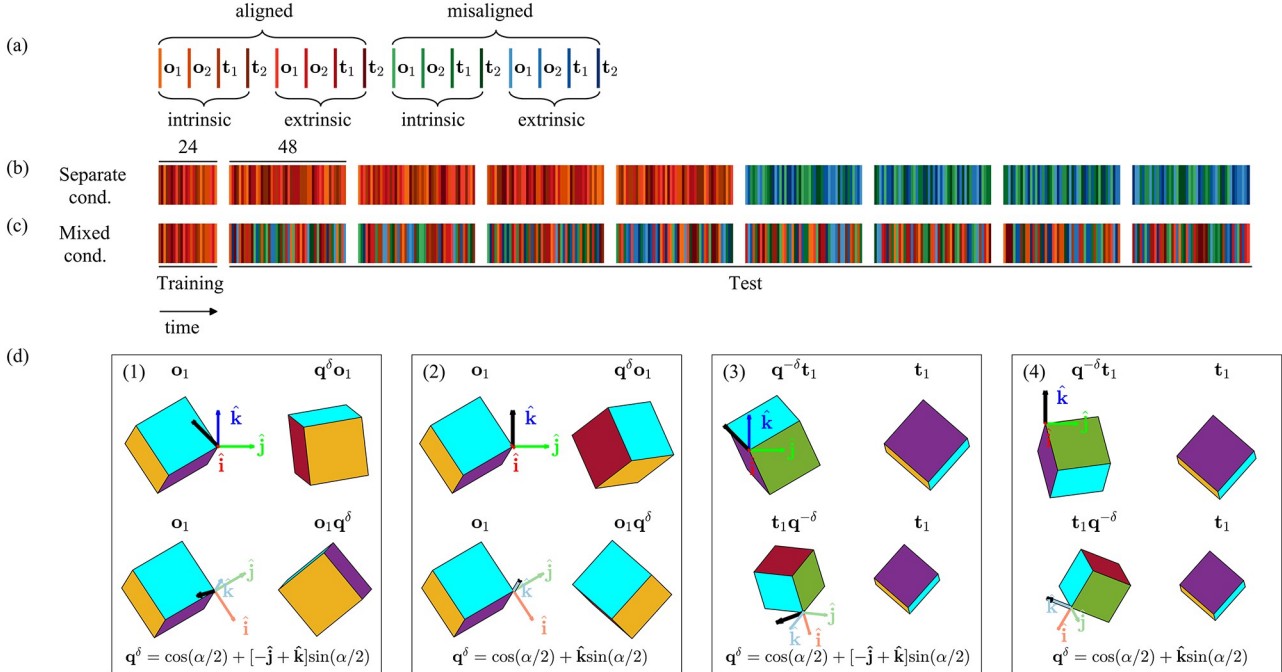

**Fig 2. Experimental design and participants groups—experiment 1.** The experiment consisted of two sessions: Training (24 trials) and Test ($48 \times 8$ trials). In the Training session, all targets were presented by rotating the initial cursor orientation around an axis that was aligned to either an intrinsic or an extrinsic reference frame. The Test session of the Separate cond. group contained four trial sets with aligned axes followed by four trial sets with misaligned axes. The Test session of the Mixed cond. group contained eight trial sets with mixed aligned and misaligned axes. In both groups, half of the trials initiated from two fixed initial hand orientations ($\mathbf{o}_1$ and $\mathbf{o}_2 \in \mathbb{H}_1$), while the rest had two fixed target orientations ($\mathbf{t}_1$ and $\mathbf{t}_2 \in \mathbb{H}_1$). (a) Legend for the different experimental conditions of each trial. (b) The order of the trials in the Separate cond. group. (c) The order of the trials in the Mixed cond. group. (d) Examples of trials from the experiment. In each box, the left column shows the initial cursor orientation and the right column shows the respective target orientation (the notations of the initial cursor and target orientations, and the transition quaternion are in the inset; see Table 1 for more details). $\alpha$ is the rotation angle, and $\mathbf{q}^{-\delta}$ is the inverse of the transition quaternion. The top row shows trials constructed using rotation axes that are either misaligned (box 1 and 3) or aligned (box 2 and 4) with an extrinsic reference frame (dark red, green, and blue arrows). The bottom row shows trials constructed using rotation axes that are either misaligned (box 1 and 3) or aligned (box 2 and 4) with an intrinsic reference frame (light red, green, and blue arrows).

addition, we rotated the cursor in two different ways: by rotating with respect to an extrinsic coordinate system (using Eq 3) and with respect to an intrinsic coordinate system (using Eq 4). We chose the initial cursor and target orientations, as well as the range of $\alpha$ (40° – 60°), to allow for reasonable rotations within the limits of the rotational workspace of the hand and the robotic handle.

For a given initial orientation, we chose rotation axes that would allow us to test for the effect of the alignment of the ideal rotation axis (i.e., the axis that one should rotate around to move along a geodesic) on the QGS. Therefore, we chose rotation axes that were aligned or misaligned with either an extrinsic or an intrinsic reference frame (see Table 1 and Fig 2d for examples of the experimental conditions). If rotations around aligned axes would result in lower QGS values than rotations around misaligned axes, this could imply that the alignment has significance in the control of orientation by the sensorimotor system. Moreover, it would hint that the sensorimotor system uses either intrinsic or extrinsic Euler angles—three consecutive rotations around key axes in intrinsic or extrinsic coordinates—to represent the orientation of rigid bodies.

The experiment consisted of two sessions: Training and Test (Fig 2a–2c). The Training session (not analyzed) consisted of one set of 24 trials. The Test session consisted of eight sets of 48 trials each. We pseudo-randomized the order of trials within each set beforehand,

and provided participants with a one minute break after each set. In the Training session, participants performed six trials starting from $\mathbf{o}_1$ and six trials starting from $\mathbf{o}_2$. To present participants with the target, we rotated the initial cursor orientation around one of three axes that were aligned to either an extrinsic or an intrinsic reference frame (see Eqs 3 and 4). In addition, participants performed six trials towards $\mathbf{t}_1$ and six trials towards $\mathbf{t}_2$. To present participants with the initial cursor orientation, we inversely rotated the targets around the same axes.

In the Test session, we split the participants into two groups of seven participants: Mixed cond. and Separate cond. In the Mixed cond. group, all sets consisted of 24 trials in which the targets were rotated around an axis aligned with either an extrinsic or an intrinsic reference frame, and another 24 trials in which the rotation axes were misaligned with either one of these frames. In the Separate cond. group, the first four sets consisted of 48 trials with axes that were aligned to either reference frame, and the remaining four sets consisted of 48 trials with axes that were misaligned with the reference frames. Throughout the Test session, each participant performed eight repetitions of each trial with the same initial and target orientations.

**Experiment 2.** The distribution of the QGS in experiment 1 led us to conclude that orientation-matching movements are generally performed by following a geodesic in $\mathbb{H}_1$. This result established a baseline for studying adaptation and generalization of rotation-based perturbation during an orientation-matching task. If we had not seen such evidence of path planning optimization, we would not have been able to compare rotational variables between non-perturbed and perturbed movements.

In this experiment, we applied a visuomotor rotation transformation between the rotation of the handle of the robotic manipulator and a three-dimensional cursor. Let $\mathbf{q}_r = \cos \theta_r/2 + \hat{\mathbf{n}}_r \sin \theta_r/2$ be the visuomotor rotation, where $\theta_r = 60°$ is the perturbation angle and $\hat{\mathbf{n}}_r = \hat{\mathbf{i}}$ is the perturbation axis. The perturbation angle was chosen such that the effects of adaptation and transfer, if observed, could be large enough compared to the natural variability of human movements [37, 38]. We used $\mathbf{q}_r$ to perturb the extrinsically represented instantaneous axis ($\hat{\mathbf{n}}^\delta$) to achieve the perturbed instantaneous axis ($\hat{\mathbf{n}}_p^\delta$) using Eq 2:

$$\hat{\mathbf{n}}_p^\delta = \mathbf{q}_r \hat{\mathbf{n}}^\delta \mathbf{q}_r^{-1}. \tag{13}$$

The perturbed instantaneous axis is generally a noisy signal and is ill-defined when the angular velocity is low. To avoid noisy visual feedback, we considered a delayed signal, such that $\mathbf{q}_i^\delta = \mathbf{q}_{i+D}\mathbf{q}_i^{-1}$, where $D = 80$ is the delay in samples. This resulted in a 20 [ms] visual delay, which is considered to be unnoticeable [48]. A sudden exposure to the perturbation causes a deviation from the desired visual scene. If one starts with initial orientation $\mathbf{q}_I$ and matches it to a target $\mathbf{q}_t = \cos \theta_t/2 + \hat{\mathbf{n}}_t \sin \theta_t/2$ by following a geodesic, then the visuomotor rotation would result in perturbed trajectories:

$$\mathbf{q}_{p,i} = \cos h_i\theta_t/2 + \mathbf{q}_r\hat{\mathbf{n}}_t\mathbf{q}_r^{-1} \sin h_i\theta_t/2, \tag{14}$$

$$\mathbf{q}_{p,i}^\delta = \cos \Delta h_i\theta_t/2 + \mathbf{q}_r\hat{\mathbf{n}}_t\mathbf{q}_r^{-1} \sin \Delta h_i\theta_t/2. \tag{15}$$

One way to compensate for the perturbation is to learn the inverse rotation ($\mathbf{q}_r^{-1}$) by constructing an internal model of the perturbation. Once built, the internal model can be used to rotate around an inversely rotated axis ($\hat{\mathbf{n}}_c = \mathbf{q}_r^{-1}\hat{\mathbf{n}}_t\mathbf{q}_r$) using a feed-forward control to achieve

the desired unperturbed visual scene ($\mathbf{q}_{up}$):

$$
\begin{aligned}
\mathbf{q}_{up,i} &= \cos h_i\theta_t/2 + \mathbf{q}_r\hat{\mathbf{n}}_c\mathbf{q}_r^{-1}\sin h_i\theta_t/2 \\
&= \cos h_i\theta_t/2 + \mathbf{q}_r(\mathbf{q}_r^{-1}\hat{\mathbf{n}}_t\mathbf{q}_r)\mathbf{q}_r^{-1}\sin h_i\theta_t/2 \qquad (16) \\
&= \cos h_i\theta_t/2 + \hat{\mathbf{n}}_t\sin h_i\theta_t/2.
\end{aligned}
$$

To test whether the inverse rotation was implemented, it is common to remove the perturbation and check for an aftereffect of adaptation while repeating the same task. Assuming the perturbation was fully compensated, a complete aftereffect will result in rotation around an inversely perturbed axis ($\mathbf{q}_{ip,i} = \cos h_i\theta_t/2 + \mathbf{q}_r^{-1}\hat{\mathbf{n}}_t\mathbf{q}_r\sin h_i\theta_t/2$). If the aftereffect is incomplete, then the hand would rotate around an axis inversely perturbed by the amount of the aftereffect. If the participant accurately aimed towards the target, then there is no aftereffect.

To probe in which representation the internal model of the perturbation is built, we removed the perturbation after a training phase. We tested for an aftereffect in the transfer of the adaptation to an orthogonal initial hand orientation—the initial orientation used during training rotated 90˚ around $\hat{\mathbf{i}}$. As in Eq 16, a transfer could be achieved by learning the inverse rotation ($\mathbf{q}_r^{-1}$), and applying it on the axis that corresponds to a geodesic starting from the new initial orientation.

Thirty right handed volunteers participated in experiment 2 (13 males and 17 females, all in the age range of 23–28). The participants were naive with regard to the purpose of the experiments and were reimbursed for their participation.

The experiment consisted of 420 trials, in which the participants were required to perform the orientation-matching task with a single, fast hand rotation. Each trial began with the appearance of the target to the left of the cursor. To present the target, we rotated the initial cursor orientation by 50˚ around an axis. We considered the movement's initiation time at the first time stamp in which the angular speed exceeded 0.25 [rad/s]. In order to avoid considering unintentional stops as the termination of the movement, we considered the first time stamp in which the angular speed dropped below 0.25 [rad/s], after exceeding 1 [rad/s], to be the movement's termination. Following each movement, we removed the target and the cursor turned red, while the robotic manipulator autonomously translated and rotated to the initial position and orientation of the subsequent trial using the same controllers as in experiment 1. In order to guide the participants to perform their movements within a desired duration range and to reduce variability in the data, we displayed feedback regarding the duration of the movement. We displayed a "Move Slower" notification if the movement lasted less than 400 [ms] and a "Move Faster" notification if it lasted more than 600 [ms]. We displayed an "Exact" notification to the side of the duration feedback if the cursor-to-target angular distance at the time of trial termination was lower than 15˚. We displayed a "Perfect" notification if both duration and accuracy conditions were satisfied. Following that, we initiated the next trial when the angular distance between the orientations of the cursor and the next initial orientation decreased below 5˚ for 100 [ms]. Then, we colored the cursor back as before (see Fig 1b), and presented participants with a new target.

The experimental protocol, inspired by [8], is summarized in Fig 3. To familiarize participants with the orientation-matching task, they performed 60 familiarization trials without any perturbation. Participants performed the first 30 trials (FML1) starting from the unrotated state of the hand ($\mathbf{o}_1 = \mathbf{q}_I$). In the next 30 trials (FML2), we rotated the initial orientation by 90˚ around $\hat{\mathbf{i}}$ ($\mathbf{o}_2 = 1/\sqrt{2} + \hat{\mathbf{i}}/\sqrt{2}$). The rest of the experiment consisted of three sessions: Baseline (BL), Training (TRN) and Transfer (TFR). In the BL session, participants performed the task for two sets of 60 trials each (BL1 and BL2) without any perturbation. Trials in BL1

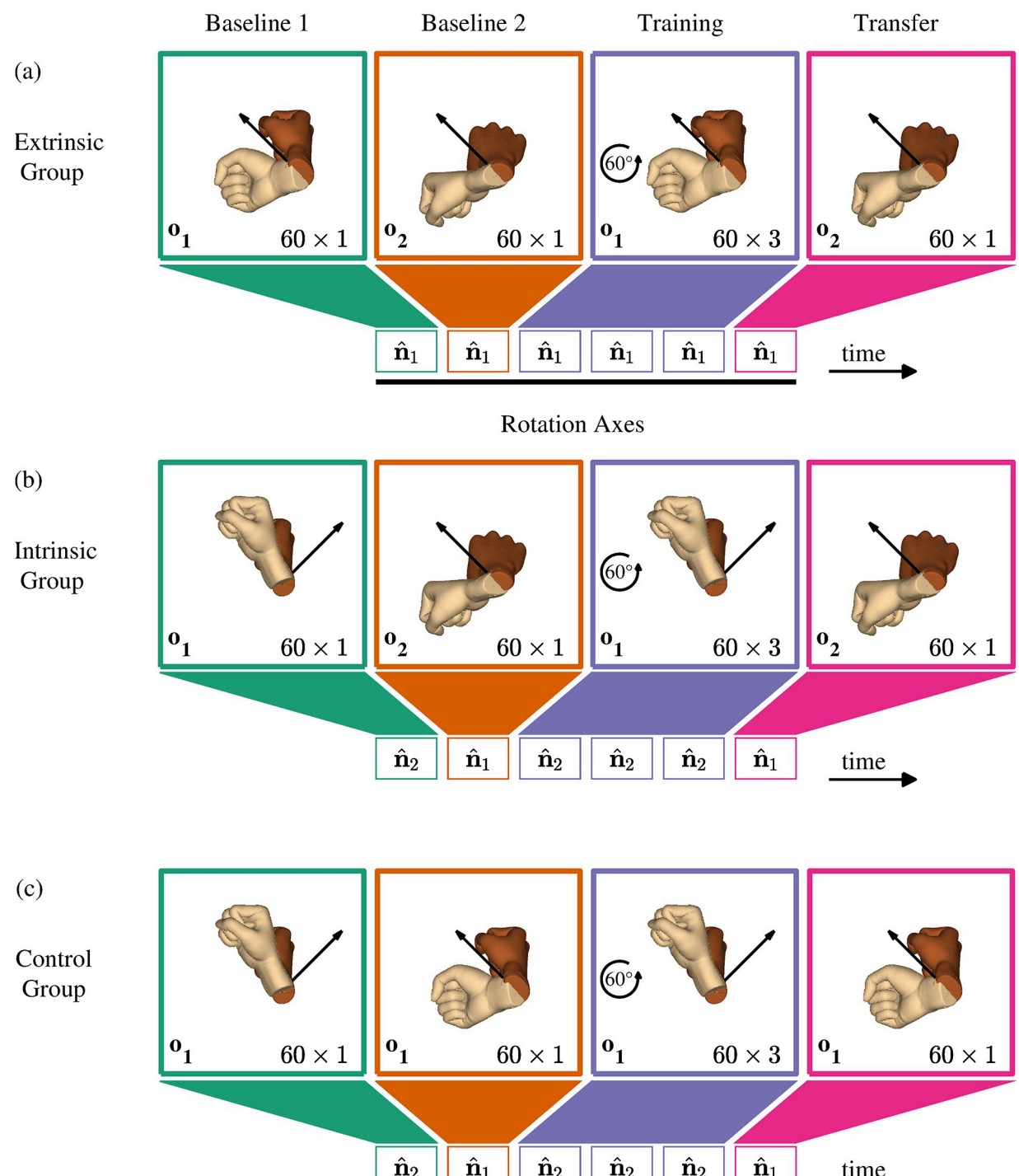

**Fig 3. Experimental design and participants groups—experiment 2.** All participants performed two sets of 60 baseline trials, three sets of 60 training trials, in which the cursor's rotation was perturbed by 60° around $\hat{\mathbf{i}}$, and one set of 60 transfer trials, in which the perturbation was removed. The black rotation axes are drawn with respect to an extrinsic reference frame. The initial cursor and target orientations in each set of trials are represented by the dark and light hands, respectively. The colors represent different sessions of BL (BL1—green, BL2—orange), TRN (blue) and TFR (pink). (a) Participants in the Extrinsic group performed all rotations towards similar targets in extrinsic coordinates—we rotated the initial orientation by 50° around $\hat{\mathbf{n}}_1 = -\hat{\mathbf{j}}/\sqrt{2} + \hat{\mathbf{k}}/\sqrt{2}$ (in extrinsic coordinates). (b) Participants in the Intrinsic group performed all rotations towards similar targets in intrinsic coordinates—we rotated the initial orientation by 50° around $\hat{\mathbf{n}}_2 = \hat{\mathbf{j}}/\sqrt{2} + \hat{\mathbf{k}}/\sqrt{2}$ (in intrinsic coordinates). In both groups, trials in BL1 and TRN started from $\mathbf{o}_1$, and trials in BL2 and TFR started from $\mathbf{o}_2$. (c) To remove the effect of initial hand orientation, participants in the Control group started all rotations from $\mathbf{o}_1$. The targets in BL1 and TRN were similar to those in BL1 of the Intrinsic group, and the targets in BL2 and TFR were similar to those in BL1 of the Extrinsic group.

started from $\mathbf{o}_1$, while trials in BL2 started from $\mathbf{o}_2$. In the TRN session, we repeatedly exposed the participants to the visuomotor rotation (as detailed in Eq 13) for three sets of 60 trials with the same initial and target orientations as in BL1. The TFR session was similar to BL2; we abruptly removed the perturbation while testing whether the learning of the perturbation was transferred to a different initial orientation. We provided participants with a one minute break after each set of trials.

To test in which coordinate system the visuomotor rotation was represented, we planned two groups of ten participants: Extrinsic and Intrinsic. In the Extrinsic group, all targets were identical in extrinsic coordinates—we rotated the initial cursor orientation by 50˚ around $\hat{\mathbf{n}}_1 = -\hat{\mathbf{j}}/\sqrt{2} + \hat{\mathbf{k}}/\sqrt{2}$ using Eq 3 (Fig 3a). In the Intrinsic group, all targets were identical in intrinsic coordinates—we rotated the initial cursor orientation by 50˚ around $\hat{\mathbf{n}}_2 = \hat{\mathbf{j}}/\sqrt{2} + \hat{\mathbf{k}}/\sqrt{2}$ using Eq 4 (Fig 3b). One can imagine an extrinsic coordinate system by fixing it on the shoulder, and an intrinsic coordinate system by placing it on the wrist, which rotates as a function of the angles of the joints. The relation between the extrinsic and intrinsic coordinate systems is the hand's inverse kinematics, which maps the position and the orientation of the wrist to the angles of the hand's joints. Therefore, if one learns in an intrinsic coordinate system, it could be attributed to learning a new joint configuration. In contrast, if one learns in an extrinsic coordinate system, it could be attributed to learning a new rotation axis in visual coordinates. We chose the target orientations such that the ideal rotation axes (i.e., the axes used when following a geodesic) would be orthogonal between the two groups in BL1 and TRN, but identical in BL2 and TFR. Therefore, a difference between groups in the transfer of adaptation to an orthogonal initial orientation could be associated with the coordinates in which the perturbation was learned. However, such difference may also stem from the difference in the training targets, regardless of the trained coordinates (see the Discussion section).

To reveal the coordinate system in which the visuomotor rotation was learned, we rotated the initial hand orientation by 90˚ around $\hat{\mathbf{i}}$ and tested for transfer of adaptation in each group. However, the ability to separate between the two coordinates requires that the generalization of the perturbation to different directions would be sufficiently narrow. For instance, a wide generalization could cause a false pretense—upon removal of the perturbation, we would observe similar rotations in both groups, but this could be attributed to transfer of adaptation to an orthogonal initial hand orientation in one group and to generalization of adaptation to an orthogonal direction in the other group [24]. Therefore, we planned a control group of ten participants to test to what extent the adaptation generalizes to an orthogonal direction without changing the initial orientation (Fig 3c).

The hypotheses for the expected hand rotation in each one of the learning coordinates are depicted in Fig 4. Following two baselines, participants were trained with the perturbation. A possible way to compensate for the visuomotor rotation is to rotate the hand around an axis rotated by the inverse of the visuomotor rotation (Fig 4, Late Training). If the perturbation was learned and generalized to an orthogonal initial hand orientation, it is expected to be transferred in at least one of the learning coordinates (Fig 4, Early Transfer), followed by a full washout (Fig 4, Late Transfer).

## Data analysis

We recorded the orientations of the handle of the robotic manipulator as rotation matrices at 1 [kHz]. To account for small changes in the sampling rate, we resampled the rotation matrices to a constant 1 [kHz] sampling rate using spherical linear interpolation. Then, we downsampled the matrices to 100 [Hz] and transformed them into quaternions. We computed the angular velocity from the quaternion trajectories and low-pass filtered it at 6 [Hz] with a 4th

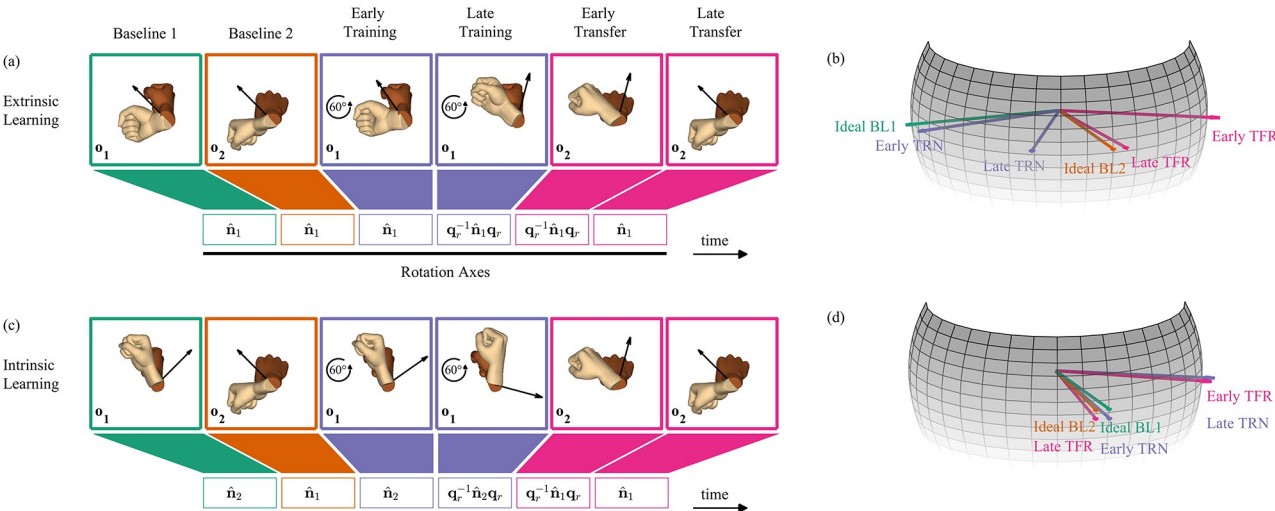

**Fig 4. Hypotheses of learning and generalization of a visuomotor rotation – the case of orientation-matching.** (a)-(b) Extrinsic learning hypothesis—transfer of learning of an extrinsically represented axis. (c)-(d) Intrinsic learning hypothesis—transfer of learning of an intrinsically represented axis. The dark and light hands represent the expected initial and final orientations of the hand, respectively. The black rotation axes are drawn with respect to an extrinsic reference frame. The colors are as in Fig 3. If participants adapt to the visuomotor rotation while training, their aim is expected to gradually deviate from the ideal BL1 axis in the direction opposite to that of the perturbation. If participants transfer the perturbation across orthogonal initial orientations, their aim is expected to deviate in the same direction from the ideal BL2 axis in the early transfer trials. The spheres present the hypotheses regarding the hand rotation axis in an intrinsic reference frame at each key trial. The early TRN axis may deviate from the ideal BL1 axis due to corrections. Axes which are ideally equal, such as the ideal BL1 and ideal BL2 axes in (d), were separated for better visualization.

order zero-lag Butterworth filter using the *filtfilt* function in MATLAB (The MathWorks, Natick, Massachusetts, USA).

We removed trials from the analysis if the participant failed to follow the instructions of the task; for example, trials in which participants used their left hand to correct or support the right hand were removed. Other examples include grasping the handle incorrectly, letting go of the handle and looking away from the three-dimensional viewer.

**Experiment 1.** A total of 37 trials were removed (0.69%), with a maximum of 9 trials per participant (2.34%). We analyzed each trial from the first time stamp in which the angular speed of the hand passed 10% of its maximum until the time stamp in which the cursor-to-target angular distance decreased below 15° for a period of 400 [ms]. We computed the QGS of each trial using Eq 11.

**Experiment 2.** A total of 35 trials were removed (0.32%), with a maximum of 8 trials per participant (2.22%). Based on the first experiment, we knew that people tend to follow a geodesic in $\mathbb{H}_1$ when asked to reorient a cursor to match a target. Therefore, we were able to quantify a single rotation aiming axis at the time of peak angular speed. To support the reliability of the adaptation analysis, we ensured that the participants did rotate their hand along a geodesic in the BL trials using the QGS values, computed from the time of trial initiation until the time of peak angular speed. In the subsequent analyses, we analyzed each trial from the first time stamp in which the angular speed exceeded 10% of its maximum. Let $\{\mathbf{q}\}_{i=1}^{N}$ be a discrete time, $N$-sample orientation trajectory of the handle of the robotic manipulator, where $\mathbf{q}_i = \cos\theta_i/2 + \hat{\mathbf{n}}_i \sin\theta_i/2$. First, to have the initial orientation equal to $\mathbf{q}_b$, we rotated each quaternion sample by $\mathbf{q}_1^{-1}$ (see the Mathematical formulation section). Then, we defined the aiming axis in each trial:

$$\hat{\mathbf{n}}_a = \arg\max_{\hat{\mathbf{n}} \in \mathbb{R}^3} \|\omega\|, \tag{17}$$

where ω is the angular velocity of the hand:

$$\omega = \frac{\theta^\delta}{\Delta t} \hat{\mathbf{n}}^\delta,$$ (18)

where $\Delta t$ is the time sampling interval.

To reduce the variability between participants, for each participant we removed the baseline of the set of aiming axes. To account for the three-dimensional information of the data, we removed the baseline on the sphere as follows. First, we defined the baseline axis, denoted by $\langle \hat{\mathbf{n}}_a \rangle$, as the average over the aiming axes of 20 randomly chosen trials out of the last 40 trials in each BL set (their normalized sum). Then, we removed the baseline by rotating all aiming axes by the rotation that brings the baseline axis to the ideal rotation axis ($\hat{\mathbf{n}}_{id}$), i.e., all aiming axes were rotated by an angle $\arccos(\langle \hat{\mathbf{n}}_a \rangle \cdot \hat{\mathbf{n}}_{id})$ around an axis $\frac{\langle \hat{\mathbf{n}}_a \rangle \times \hat{\mathbf{n}}_{id}}{\|\langle \hat{\mathbf{n}}_a \rangle \times \hat{\mathbf{n}}_{id}\|}$.

In addition to the spherical data analysis, we quantified the aiming angle—a scalar value that estimates the direction of rotation. We defined it as the elevation angle of the aiming axis from a plane spanned by the perturbation axis ($\hat{\mathbf{n}}_r$) and the ideal rotation axis ($\hat{\mathbf{n}}_{id}$); the latter is different between different experimental sessions:

$$\text{Aiming angle} = \arctan 2\left( \hat{\mathbf{n}}_a \cdot (\hat{\mathbf{n}}_r \times \hat{\mathbf{n}}_{id}), \sqrt{(\hat{\mathbf{n}}_a \cdot \hat{\mathbf{n}}_r)^2 + (\hat{\mathbf{n}}_a \cdot \hat{\mathbf{n}}_{id})^2} \right).$$ (19)

The aiming angle spans between $-\pi/2$ and $\pi/2$, and it is zero when $\hat{\mathbf{n}}_a = \hat{\mathbf{n}}_{id}$. This is similar to the aiming angle in studies of two-dimensional translations, that is, the angular distance between the aiming axis and the straight line that connects between the initial position and the target.

## Statistical analysis

**Experiment 1.** The statistical analyses were performed using a custom-written MATLAB code. The code is provided at https://github.com/Bio-Medical-Robotics-BGU/Orientation-control-strategies. We used bootstrap to compute the 95% confidence interval (CI) for the median QGS of each participant. The median was chosen as the parameter for central tendency since the scores have a skewed distribution (a normal distribution was rejected using Lilliefors test with p = 0.001). In the orientation-matching task, we considered a deviation from the minimal angular displacement required to match the cursor to the target to be small if it was less than 10°. Therefore, since the average angular displacement between the initial cursor and target orientations was 50°, we considered a rotation as geodetic enough if its QGS was less than 1.2.

To check whether participants changed their QGS throughout the experiment, we conducted a Wilcoxon rank sum test for the equality of the medians of the geodicity scores of the first 10 trials and the last 10 trials of the Test session. To check for large effects, we computed the median of the absolute QGS differences over participants in each group.

To determine significant effects of trial conditions on the QGS, we conducted a six-way repeated measures ANOVA test with the QGS as the response variable—one between the participants' factor of the group (Mixed cond., Separate cond.) and five within the participants' factors of initial hand orientation ($\mathbf{o}_1$, $\mathbf{o}_2$), target orientation ($\mathbf{t}_1$, $\mathbf{t}_2$), axis coordinate system (extrinsic, intrinsic), axis alignment (aligned, misaligned) and rotation axis (see Table 1). We checked for the main effect, as well as for first order interactions. Prior to the analysis, the scores were transformed using $\log_{10}$ to reduce deviation from normality. Significant effects were defined as those with a probability level of $p < 0.05$ (For convenience, when used in figures and tables, $p < 0.05$ is notated as $^*$ and $p < 0.001$ is notated as $^{**}$). The effect size is

reported using partial eta-squared ($\eta_p^2$) for the ANOVA test. Since none of the factors were found to be large nor significant, we did not apply further multiple comparisons within and between factors.

**Experiment 2.** To ensure that the BL trials were geodetic enough, and that their geodicity was stable, we computed the median QGS value for each group, as well as the 95% bootstrap CI for the median QGS, and the 5% and 95% percentiles of the QGS.

We used two statistical approaches to quantify learning and generalization effects. In both approaches we analyzed key stages in the experiment, including late BL1 and BL2 trials, early and late TRN trials, and early and late TFR trials. A visual examination of the adaptation curves implied that the learning was fast, but the transfer was slow. Fast processes are easier to observe when using a single-trial analysis (i.e., considering only the first trial of TRN as representing the effect of the initial exposure to the perturbation) instead of a multiple-trials analysis (i.e., using the first few trials of TRN to characterize the effect of the initial exposure). Slow processes, on the other hand, are still visible when using multiple-trials analysis. However, using a single trial-analysis is prone to be noisy, as opposed to multiple-trials analysis. To be able to observe both fast and slow processes, we conducted a sensitivity analysis on the number of trials averaged in the statistical analysis, i.e., we compared within and between groups each time by averaging over a different number of trials (1–10). For example, when using three trials in the analysis, we averaged over trials 118–120 (late BL1), as well as trials 178–180 (late BL2), trials 181–183 (early TRN), trials 358–360 (late TRN), trials 361–363 (early TFR), and trials 418–420 (late TFR). We report the statistical results of a three-trials analysis, as it is able to capture large effects of both learning and transfer behaviour. However, we provide all the results from the sensitivity analysis in S1 Appendix.

To test for the effect of initial exposure to the perturbation, we compared between the late BL1 and early TRN trials. To test if the participants fully adapted to the perturbation, we compared between the late BL1 and late TRN trials. To quantify the extent of adaptation, we compared between the early and late TRN trials. To test for transfer of adaptation, we compared between the late BL2 and early TFR trials. To test if the transfer was fully washed out, we compared between the late BL2 and late TFR trials, as well as the early and late TFR trials. Then, we compared within key trials between groups. We tested for differences in the early TRN trials and the late TRN trials. Furthermore, to reveal the coordinates of learning, we compared between the Intrinsic and Extrinsic groups in the early TFR trials.

In the spherical data analysis, we used the 'Directional' package in R [49] to analyze differences in the baseline corrected aiming axes. We assumed that the axes are sampled from a symmetric distribution around a mean direction ($\boldsymbol{\mu}$) with a concentration parameter ($\kappa$). To test for the equality of mean directions of two samples of aiming axes, we used a non-equal concentration parameters approach for a one-way spherical ANOVA test. Additionally, to test whether an axis could be considered the mean of a sample, we used a log-likelihood ratio test with bootstrap calibration. Significant effects were defined as those with a probability level of $p < 0.05$. We also report the angular distance between the average axes of the compared samples as a measure of the effect size. We consider a mean difference to be large if it is greater than 15°, as this was also the threshold of the cursor-to-target angular distance that awarded participants with a positive feedback on the accuracy of their rotation.

In the scalar aiming angle analysis, we quantified adaptation and transfer of learning by comparing aiming angles of the six key stages. The aiming angle is a non-periodic scalar variable that does not require any spherical treatment. Therefore, we performed a two-way mixed model repeated measures ANOVA test with the aiming angle as the response variable, one between the participants' factor of the group (Extrinsic, Intrinsic, and Control) and one within

the participants' factor of the stage (late BL1, late BL2, early TRN, late TRN, early TFR, and late TFR). When significant effects were found, we performed the planned comparisons using a t test with Bonferroni's correction. The effect size is reported using partial eta-squared ($\eta_p^2$) for the ANOVA test and Cohen's d for all t tests.

## Results

### Experiment 1

In the orientation-matching experiment, we studied whether participants performed geodetic hand rotations in quaternion space. Participants were requested to match the orientation of a three-dimensional virtual cursor to that of an oriented target using a robotic handle. We quantified the geodicity of the movement using the QGS. Fig 5 depicts the results of two trials with the same initial cursor orientation and target orientation—one geodetic rotation (Fig 5a) and one non-geodetic rotation (Fig 5b). Presented are the paths of an intrinsic reference frame that is attached to the robotic handle, the angular velocity profiles used to follow these paths, as well as its norm, and the instantaneous axis paths ($\hat{\mathbf{n}}^\delta$). In the intrinsic frame path, the circles enclose caps with angular apertures of 5˚ and 15˚ that define the accepted range for the hand's orientations at the initiation and the termination of the movement (see the Procedure and protocol section for details). While both participants matched the cursor to the target with sufficient proximity, the geodesic (QGS = 1.08) followed a shorter path compared to the non-geodesic (QGS = 4.2), rotated more smoothly and its instantaneous axis varied less.

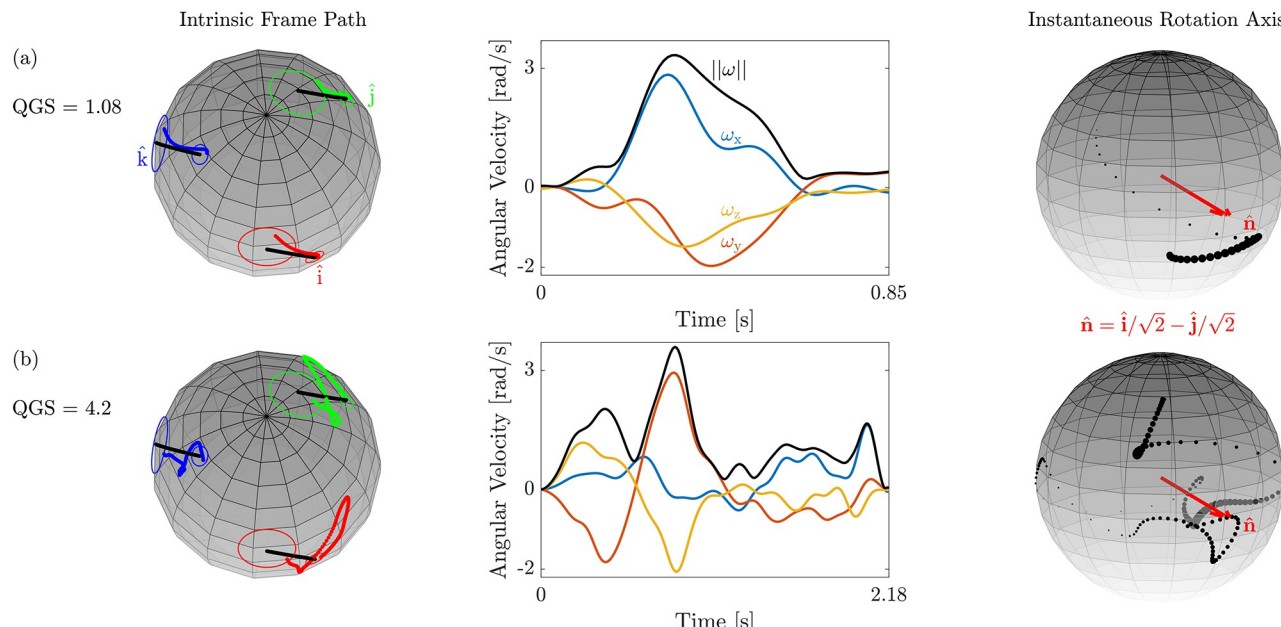

**Fig 5. Examples of a geodesic and a non-geodesic.** The left column shows the path of the intrinsic reference frame that is attached to the robotic handle ($\hat{\mathbf{i}}$—red, $\hat{\mathbf{j}}$—green, and $\hat{\mathbf{k}}$—blue). The circles enclose caps that define the accepted range for the hand's orientation at the initiation of the movement (small caps with angular aperture of 5˚) and at its termination (large caps with angular aperture of 15˚). The black curves describe the geodesic between the initial cursor orientation and the target orientation. The middle column shows the angular velocity profile (x—blue, y—orange, and z—yellow) used to follow the orientation path, as well as its norm (black), and the right column describes the path of the instantaneous rotation axis (black dots). To emphasize fast segments, the size of the black dots is scaled in proportion to the angular speed. The ideal rotation axis is shown in red. (a) The trajectory of the intrinsic frame of a geodesic follows a short path on the sphere and is characterized by a smooth angular velocity profile and a concentrated instantaneous axis path. (b) The trajectory of the intrinsic frame of a non-geodesic follows a longer path on the sphere compared to that of the geodesic and is characterized by a fragmented angular velocity profile and a dispersed instantaneous axis path. In both examples, the initial cursor orientations and the target orientations were identical.

**Participants tended to follow a geodesic in quaternion space.** The median was chosen to account for the central tendency of the QGS since its distribution is skewed (skewness = 4.57, bootstrap 95% CI = [3.16, 6.32]). When asked to match the orientation of the cursor to the target (Fig 6), i.e., their movements mostly resembled Fig 5a rather than Fig 5b. Only four participants had scores with a median 95% CI above the 1.2 threshold that we set for geodesics. Out of the rest, the CIs

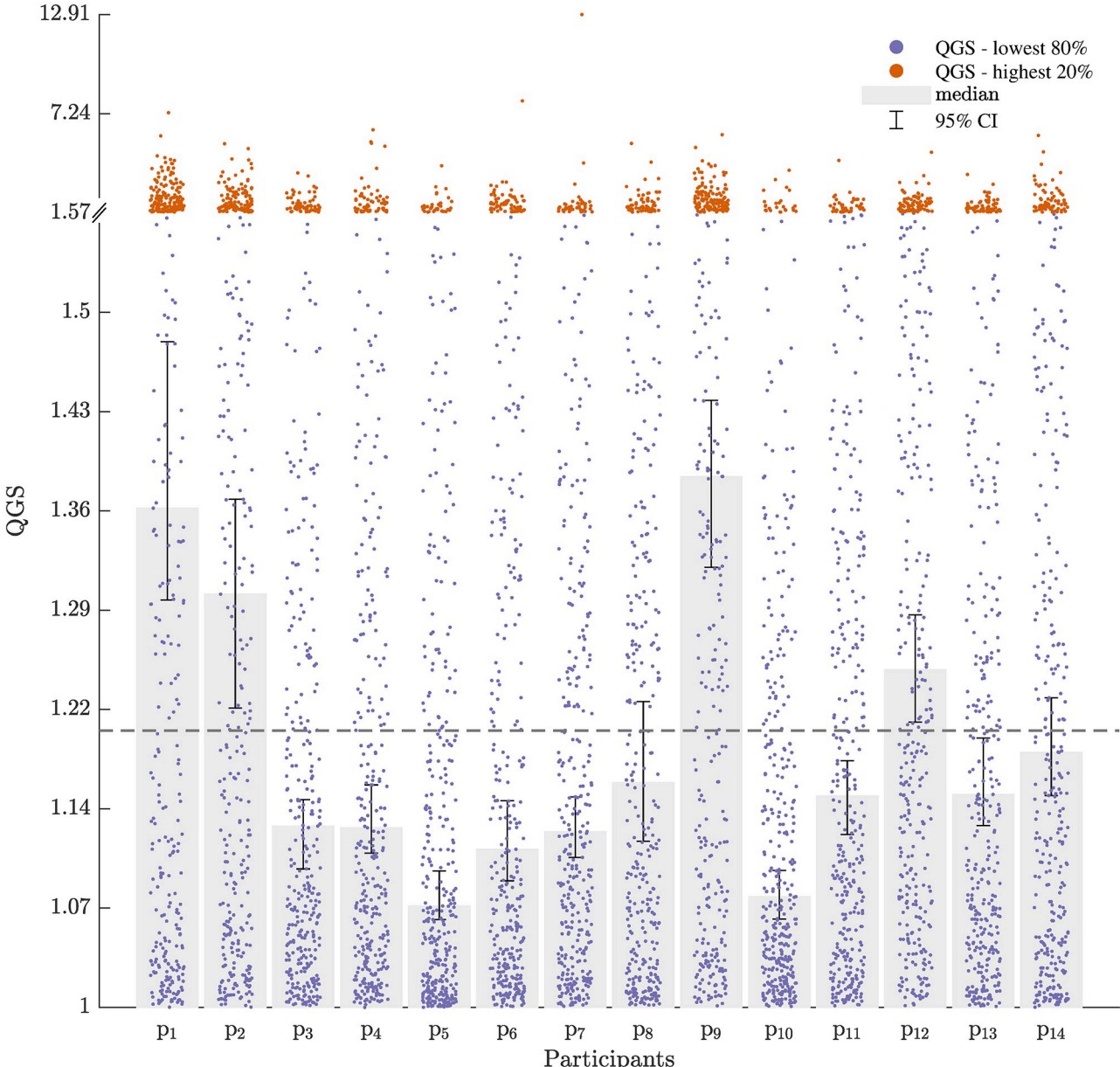

**Fig 6. Median QGS and 95% bootstrap CI.** Most of the participants performed geodetic hand rotations as indicated by the low 95% CI of the median QGS for eight of the fourteen participants ($p_3$—$p_7$, $p_{10}$, $p_{11}$ and $p_{13}$). Yet, four participants deviated largely from geodesics ($p_1$, $p_2$, $p_9$ and $p_{12}$), while the rest gave indecisive results ($p_8$ and $p_{14}$). For visualization purpose, the lowest 80% of all scores are scattered in blue along 80% of the vertical axis, and the highest 20% of all scores are scattered in orange along the remaining 20% of the vertical axis. While both ranges are linearly scaled, the gap between values is different. The gray bars start from QGS = 1 (the lower limit of the region of interest), and end at the median of the scores of all trials for each participant. The error bars mark the bootstrap 95% CI for the median. The dashed gray line at QGS = 1.2 is the upper limit of the region of interest.

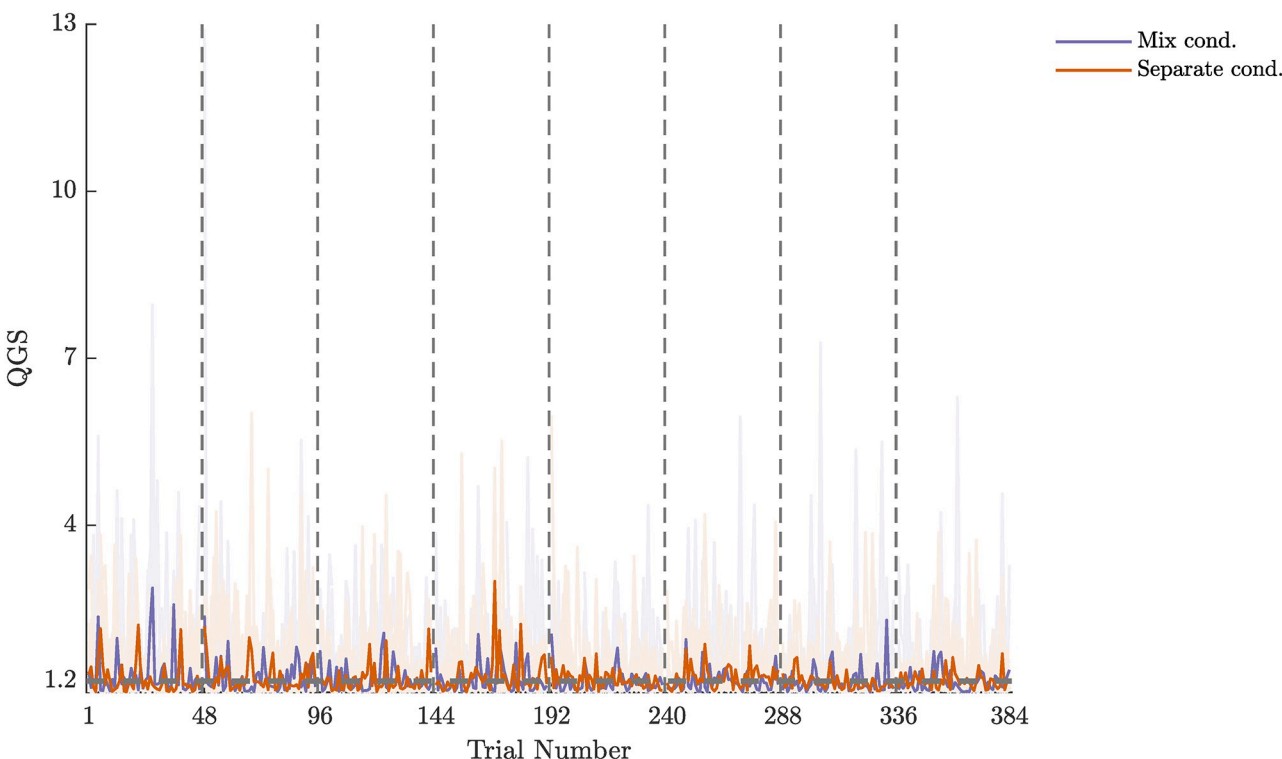

**Fig 7. The time course of the QGS values.** The QGS values were not largely reduced from the beginning of the experiment to its end. The light shaded curves show the time courses of the QGS values of each participant, and the bold curves show the time course of the median QGS in each group. The horizontal dashed gray line at QGS = 1.2 is the upper limit of the region of interest. The vertical dashed gray lines separate between the different sets of trials.

of eight participants fell below 1.2, as they mostly performed geodetic rotations and the CIs of two were inconclusive. The fact that most rotations followed geodesics suggests that hand rotations are centrally controlled, similarly to translations, possibly by optimizing geometrical properties of the orientation path. Yet, this control may not be perfect, as indicated by the large variability of the QGS, as opposed to low variability observed in planar reaching [38].

**The QGS was not largely reduced throughout the experiment.** Fig 7 shows the time course of the QGS values in the Separate cond. group and in the Mixed cond. group. We tested whether participants changed the geodicity of their rotation in median throughout the Test session using a Wilcoxon rank sum test. Participants in both groups did not largely reduce their QGS (Mixed cond.: p = 0.9, mean diff. = 0.02, Separate cond.: p = 0.07, mean diff. = 0.29). This result was expected, as the orientation-matching task is a natural motor action that is performed on a regular basis.

**The QGS is robust to changes in the conditions of the task.** A six-way repeated measures ANOVA test was used to find the effects of the experimental factors on the $\log_{10}$-transformed QGS. The statistical analysis did not reveal a significant nor large main effect of the group ($F_{1,1} = 0.96$, p = 0.51, $\eta_p^2 = 0.49$), the initial hand orientation ($F_{2,2} = 0.88$, p = 0.53, $\eta_p^2 = 0.47$), the target orientation ($F_{2,2} = 16.97$, p = 0.06, $\eta_p^2 = 0.94$), the alignment ($F_{1,1} = 0.11$, p = 0.79, $\eta_p^2 = 0.1$), the axis coordinate system ($F_{1,1} = 3.17$, p = 0.33, $\eta_p^2 = 0.76$), and the rotation axis ($F_{5,5} = 1.71$, p = 0.29, $\eta_p^2 = 0.63$). Additionally, we did not find large nor significant effects of the interaction between the group and the initial hand orientation ($F_{2,2} = 3.52$, p = 0.22, $\eta_p^2 = 0.78$), the group and the target orientation ($F_{2,2} = 0.14$, p = 0.88, $\eta_p^2 = 0.12$), the group and

the alignment ($F_{1,1}$ = 0.31, p = 0.68, $\eta_p^2$ = 0.23), the group and the axis coordinate system ($F_{1,1}$ = 0.06, p = 0.84, $\eta_p^2$ = 0.06), and the group and the rotation axis ($F_{5,5}$ = 1.03, p = 0.49, $\eta_p^2$ = 0.51). The fact that the QGS was not largely affected by any of these factors suggests that it reflects a robust motor invariant that is controlled by the sensorimotor system. Moreover, the lack of strong reliance of the QGS on the alignment with either intrinsic or extrinsic reference frames does not suggest that the orientation of a rigid body is centrally represented using Euler angles.

One might expect the large QGS trials to be a result of the transition between rotating around axes that are aligned or misaligned with an extrinsic reference frame and axes that are aligned or misaligned with an intrinsic reference frame. A visual examination of the data showed that the large QGS trials are not more common when such interference occurs, which reinforces the robustness of the QGS to the conditions of the trials.

## Experiment 2

In the visuomotor rotation adaptation experiment, we studied adaptation and generalization of a visual perturbation applied to the rotation of the cursor during an orientation-matching task. Following two baseline sets with veridical visual feedback and orthogonal initial hand orientations, participants were exposed to a remapping of the rotation of their hand by a 60° rotation. Then, by removing the perturbation and changing the initial hand orientation, we tested if participants learned in extrinsic or intrinsic coordinates. Fig 8 depicts the aiming axes and angles of an individual participant in the Extrinsic group. The participant had adapted by the

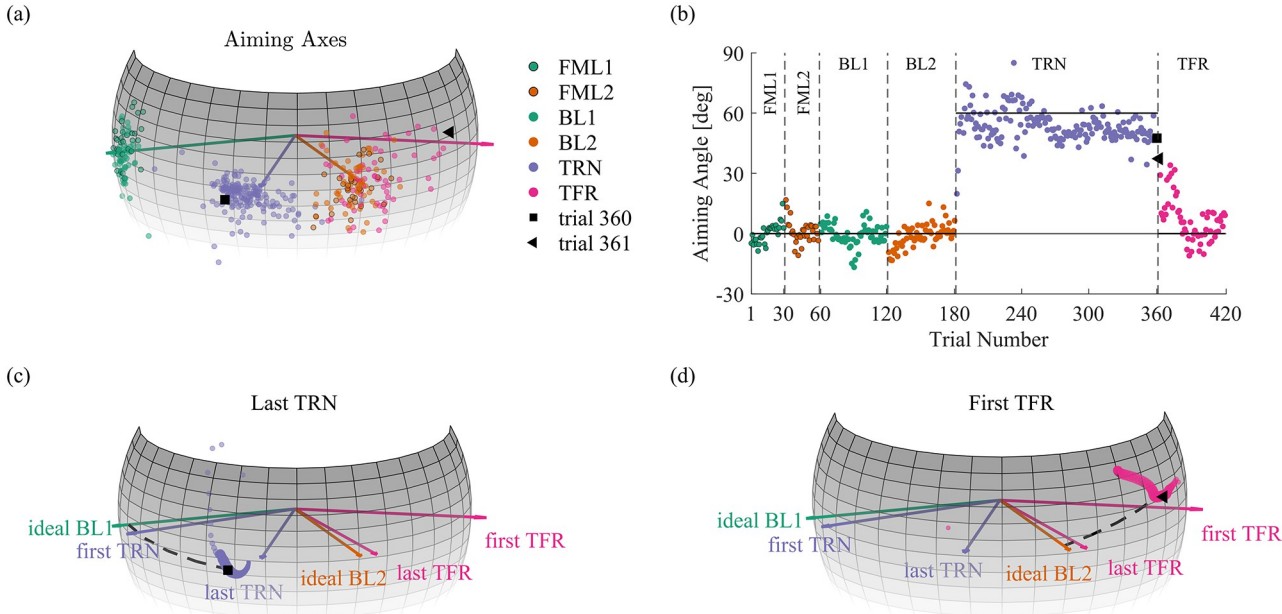

**Fig 8. Aiming axes and angles of a single participant in the Extrinsic group.** The colors are as in Fig 3. The FML1 trials are colored as The BL1 trials and enclosed by black circles, and the FML2 trials are colored as The BL2 trials and enclosed by black circles. (a) The course of the aiming axes. The last TRN axis and the first TFR axis are marked as a square and a triangle, respectively. (b) The time course of the aiming angles. The dashed lines separate between the different sets of trials, and the continuous lines indicate the ideal aiming angle at the BL session and the aiming angle required for full adaptation at the TRN session. (c) The rotation axes of the quaternion path in the last TRN trial. The extent of adaptation is the length of the arc that connects between the ideal BL1 axis and the aiming axis (i.e., the quaternion axis at the time of maximal angular speed, square). The arc reaches near the axis that represents full adaptation, which indicates that the participant had largely adapted. (d) The rotation axes of the quaternion path in the first TFR trial. The extent of transfer is the length of the arc that connects between the ideal BL2 axis and the aiming axis (triangle). The arc reaches near the axis that represents full transfer, which indicates that the participant had fully transferred the adaptation to a new initial orientation. To emphasize fast rotation samples, the size of the dots is scaled in proportion to the angular speed in (c) and (d).

end of the TRN session and transferred this adaptation to the orthogonal initial orientation when the perturbation was removed.

In the remainder of this section, all aiming axes are written and drawn in an intrinsic reference frame. Fig 9 shows the comparisons in the spherical analysis within (a-f) and between (g) groups using the three-trials average of the aiming axes of each participant. To complement the spatial spherical analysis with spatio-temporal characteristics of the rigid body's rotation, we present the intrinsically represented three-trials averaged, angular velocities of each participant in each key stage in Fig 10. To make the comparison feasible, we normalized the time to be between 0 and 1, as well as normalized the angular velocity of each participant, such that the scalar speed (i.e., the norm of the angular speed) is between 0 and 1. Fig 11a depicts the course of the aiming axes averaged across the ten participants in each group.

**Participants performed geodetic rotations in the BL session.** The reliability of the adaptation analysis highly depends on the geodicity of the rotations in the BL session. In all groups, the median QGS of all trials and participants was near 1 and had narrow CI (Extrinsic: median = 1.0081 [1.0074,1.0092], Intrinsic: median = 1.0075 [1.0069,1.0082], Control: median = 1.0067 [1.0062,1.0074]). Moreover, the QGS was stable and narrowly distributed as indicated by the 5% and 95% percentiles (Extrinsic: [1.0011,1.1637], Intrinsic: [1.0013,1.1621], Control: [1.0011,1.0914]). Therefore, the subsequent analysis is reliable.

**Participants partially adapted to the perturbation.** To check for the effect of the initial exposure to the perturbation, we compared the aiming axes of the late BL1 trials with those of the early TRN trials (Fig 9a). Since the aiming axis does not reflect an error, we did not expect it to immediately change. Nevertheless, since we provided participants with continuous visual feedback, the appearance of the perturbation could have caused them to correct their movement, rather than rotate along a geodesic, which affects the aiming axis. Moreover, averaging over three axes is expected to result in large difference between the late BL1 stage and the early TRN stage if the learning is fast. In all groups, we found a large difference (Extrinsic: $\chi^2_2 = 37.06$, p <0.001, mean diff. = 22.5˚, Intrinsic: $\chi^2_2 = 5.54$, p = 0.06, mean diff. = 15.1˚, Control: $\chi^2_2 = 3.86$, p <0.001, mean diff. = 19.4˚). However, this is likely due to the fast learning in the Intrinsic and Control groups, as a single-trial analysis revealed only small differences (Intrinsic: $\chi^2_2 = 1.3$, p = 0.52, mean diff. = 9.4˚, Control: $\chi^2_2 = 1.2$, p = 0.55, mean diff. = 7.1˚). The deviation from the late BL1 aiming axes in the initial exposure to the perturbation of participants in the Extrinsic group can be attributed to both fast learning and correction attempts, as a single-trial analysis revealed a large difference ($\chi^2_2 = 17.9$, p < 0.001, mean diff. = 15.1˚), similar to the three-trials analysis. The three-trials averaged, time and amplitude normalized, angular velocities clearly reveals the difference in the aiming axes, as the ratio of the x, y, and z coordinates differs between the late BL1 trials and the early TRN trials in all groups. They also show a fragmented rotation compared to the smooth bell-shaped velocity profile in the late BL1 trials.

Another way to test the effect of the initial exposure is to check whether the ideal BL1 axis could be considered as the mean of the aiming axes in the early TRN trials. In accordance with the former result, we found a larger effect in the Extrinsic group than in the Intrinsic group and the Control group (Extrinsic: $\boldsymbol{\mu} = 0.03\hat{\mathbf{i}} - 0.32\hat{\mathbf{j}} + 0.94\hat{\mathbf{k}}$, $\kappa = 28.27$, p <0.05, mean diff. = 26.61˚, Intrinsic: $\boldsymbol{\mu} = -0.04\hat{\mathbf{i}} + 0.85\hat{\mathbf{j}} + 0.52\hat{\mathbf{k}}$, $\kappa = 10.21$, p = 0.13, mean diff. = 13.55˚, Control: $\boldsymbol{\mu} = 0.03\hat{\mathbf{i}} + 0.89\hat{\mathbf{j}} + 0.45\hat{\mathbf{k}}$, $\kappa = 47.32$, p <0.05, mean diff. = 18.36˚). Therefore, we conclude that the participants mostly did not adapt in their early attempts with the visuomotor rotation as expected, although some showed signs of correction attempts.

To check if the participants adapted, we tested whether and how they changed the aiming axis from early to the late TRN trials (Fig 9b). In motor learning of visuomotor rotation, adaptation is often associated with movement in the direction opposite to the direction of the

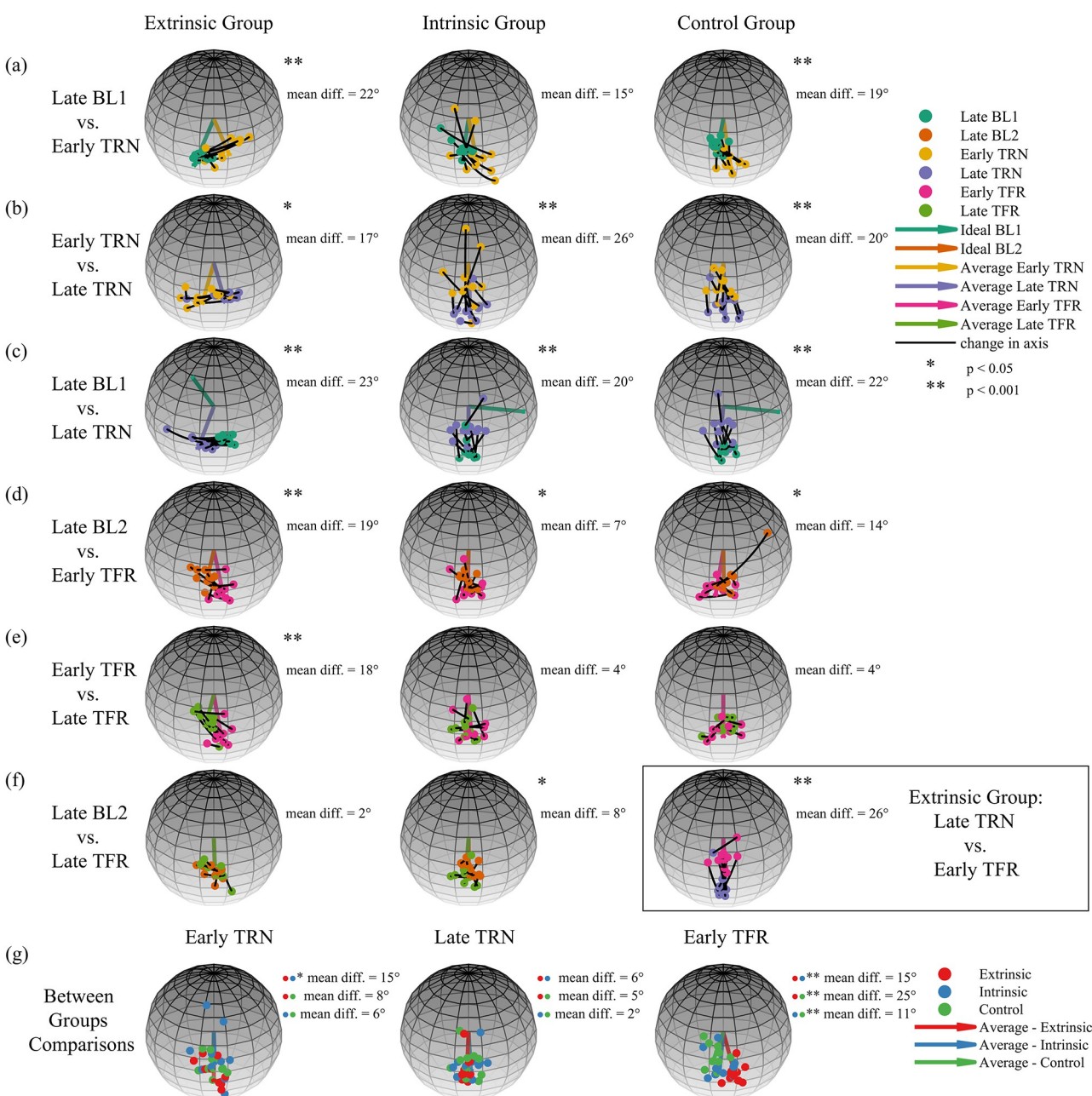

**Fig 9. Within and between groups spherical analysis.** The spheres show the comparisons of the three-trials analysis within groups between key stages (a)-(f), and between groups within key stages (g). The dots show the average axis of each participant, and the arrows depict either the mean of the sample or the ideal axis (see legend). The view of each sphere was adjusted for visibility. The within group analysis shows an effect of the initial exposure to the perturbation which may stem from correction attempts or from using the three-trials analysis (a). Furthermore, it shows a clear adaptation throughout the TRN session (b)-(c) and a transfer of adaptation only in the Extrinsic group (d), which is washed out by the end of the TFR session (e, left)-(f, left). However, the transfer is only partial (f, right). The between group analysis shows a large difference in the initial exposure to the perturbation between the Extrinsic group and both the Intrinsic and control groups (g, left). It also shows no difference in the late TRN axes, indicating similar adaptation patterns (g, middle). Most importantly, it highlights the difference in the early TFR stage between the Extrinsic and Intrinsic groups (g, right).

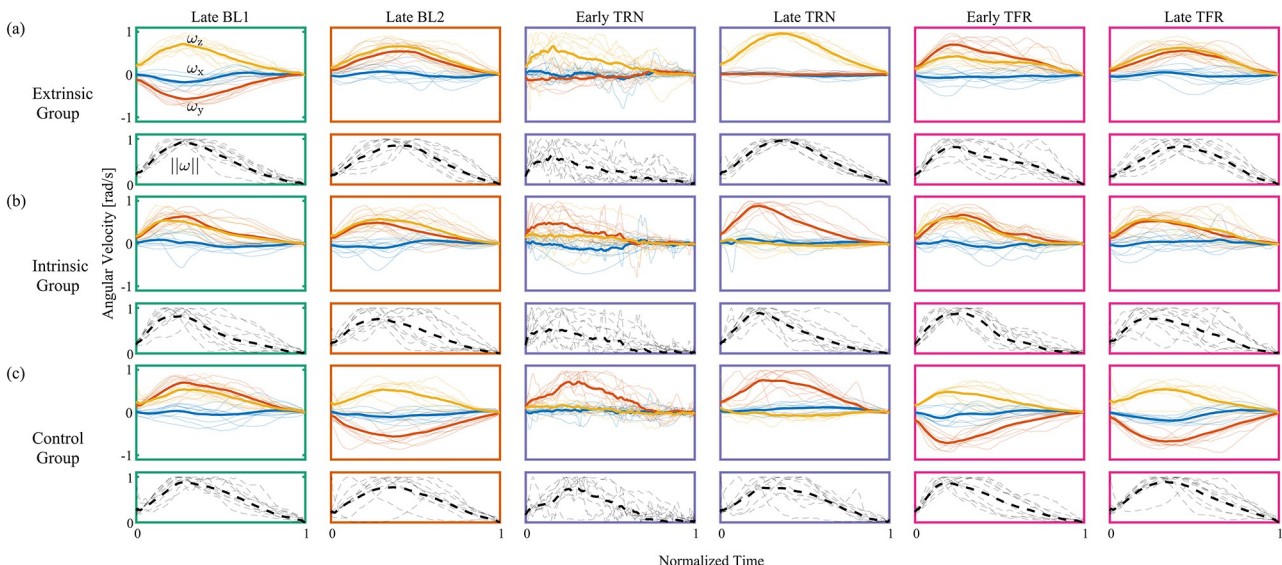

**Fig 10. Comparison of angular velocity profiles at key stages.** The solid shaded curves depict the three-trials averaged, time and amplitude normalized, angular velocity profile of each participant in key stages in intrinsic coordinates (x—blue, y—orange, and z—yellow). The solid bold curves depict the time and amplitude normalized angular velocity profiles averaged across participants. The dashed gray and black curves depict the individual and average angular speed, respectively. (a) Extrinsic group, (b) Intrinsic group, and (c) Control group. In each group, the upper panel depicts the velocity and the lower panel depicts the scalar speed. The velocity analysis complements the spherical axis analysis, as the ratio between the coordinates of the velocity profiles (i.e., the normalized vector) is the aiming axis.

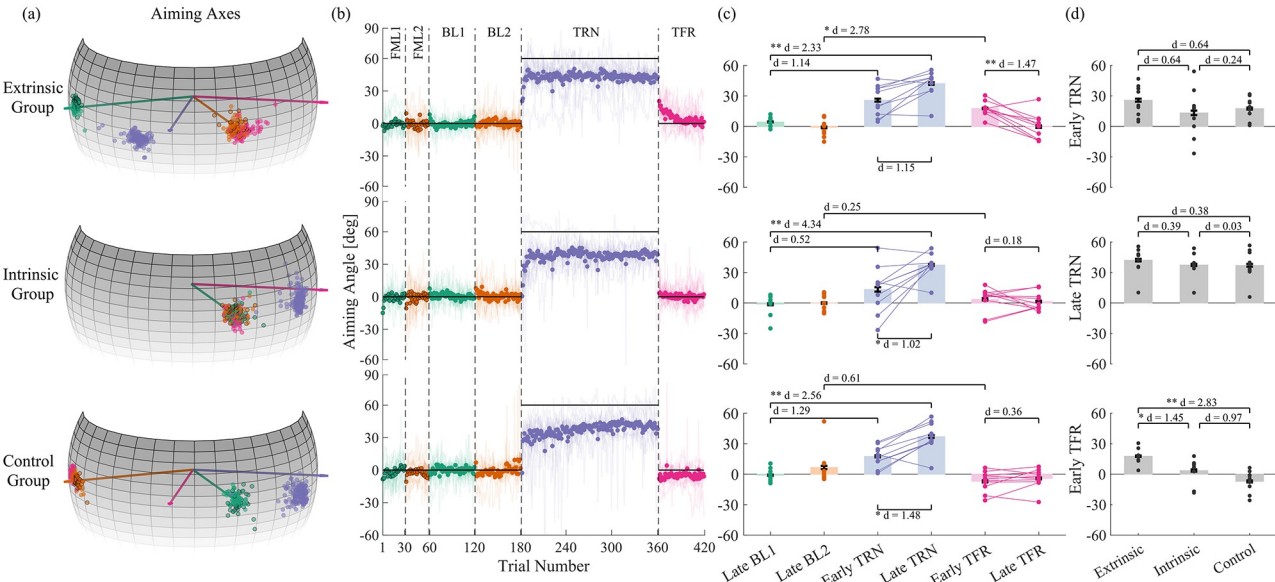

**Fig 11. Average aiming axes and angles of all groups in experiment 2 and comparisons of the aiming angles between and within groups.** The average aiming axes and angles are consistent with a representation of the partially learned perturbation in extrinsic coordinates and with a narrow generalization to different targets. The colors are as in Fig 3. (a) The course of the aiming axes (dots) averaged over participants in each group, and the ideal axis (arrows) in each session. (b) The time course of the aiming angles (dots) averaged over participants and the standard error (dark shading) of the aiming angles in each group. The light curves show the time courses of the aiming angles of each participant. The dashed lines separate between the different sets of trials. (c) Within groups comparisons of the three-trials averaged aiming angles of all participants (dots) in each of the key stages. The bars and error bars are the mean ± standard error. *p < 0.05, **p < 0.001. The effect sizes are in the inset. (d) Between groups comparisons of the three-trials averaged aiming angles of all participants in the early TRN stage, the late TRN stage, and the early TFR stage).

perturbation. Accordingly, we observed a similar effect that indicates adaptation (Extrinsic: $\chi_2^2 = 13.71$, p < 0.05, mean diff = 16.5˚, Intrinsic: $\chi_2^2 = 13.9$, p < 0.001, mean diff = 26.2˚, Control: $\chi_2^2 = 20.77$, p < 0.001, mean diff = 19.8˚).

Yet, the participants did not fully adapt. To show that, we rotated the aiming axes of the late BL1 trial by -60˚ around the perturbation axis ($\hat{\mathbf{i}}$) and found that they were significantly different from those of the late TRN trials (Fig 9c, Extrinsic: $\chi_2^2 = 52.44$, p <0.001, mean diff. = 22.7˚, Intrinsic: $\chi_2^2 = 25.59$, p < 0.001, mean diff. = 20.4˚, Control: $\chi_2^2 = 31.63$, p <0.001, mean diff. = 21.5˚). Moreover, we found a large difference between the mean of the aiming axes in the late TRN trials and the ideal BL1 axis rotated -60˚ around $\hat{\mathbf{i}}$ (Extrinsic: $\boldsymbol{\mu} = 0.11\hat{\mathbf{i}}—0.04\hat{\mathbf{j}} + 0.99\hat{\mathbf{k}}$, $\kappa = 41.65$, p <0.05, mean diff. = 18.41˚, Intrinsic: $\boldsymbol{\mu} = 0.05\hat{\mathbf{i}}+0.99\hat{\mathbf{j}} + 0.11\hat{\mathbf{k}}$, $\kappa = 31.5$, p <0.05, mean diff. = 21.89˚, Control: $\boldsymbol{\mu} = 0.09\hat{\mathbf{i}} + 0.99\hat{\mathbf{j}} + 0.12\hat{\mathbf{k}}$, $\kappa = 28.62$, p <0.05, mean diff. = 22.64˚). In fact, by the end of the TRN session, participants in the Extrinsic group rotated on average around an axis near the flexion-extension axis ($\hat{\mathbf{k}}$, Fig 10a, Late TRN), while participants in the Intrinsic and Control groups rotated on average around an axis near the radial-ulnar axis ($\hat{\mathbf{j}}$, Fig 10b, Late TRN). This ∼90˚ difference in the aiming axes between the Extrinsic group and the Intrinsic and Control groups is expected because the initial orientations were orthogonal as well.

**The perturbation was generalized to an orthogonal initial hand orientation when participants trained and transferred towards the same target in extrinsic coordinates.** To check for the generalization of the perturbation, we changed the initial orientation of the hand in the Extrinsic and Intrinsic groups and removed the perturbation. We designed the TFR trials such that participants in both groups were instructed to perform the task from the same initial orientation towards the same target. Thus, a different behaviour upon removal of the perturbation could reveal the coordinates in which the perturbation was represented in the process of adaptation. To check whether the adaptation was transferred, we compared the aiming axes in late BL2 trials and the early TFR trials. If participants did transfer the adaptation to some extent, then upon removal of the perturbation we expected them to change their aiming axis compared to late BL2 trials in the direction opposite to the direction of the perturbation. This was indeed the case for participants in the Extrinsic group, which showed a large effect of transfer (Fig 9d (left), $\chi_2^2 = 52.46$, p < 0.001, mean diff. = 19.3˚), as opposed to participants in the Intrinsic group who showed only small, yet statistically significant, transfer (Fig 9d (middle), $\chi_2^2 = 7.06$, p <0.05, mean diff. = 6.6˚). Accordingly, we observed a large difference between the ideal axis of BL2 and the mean of the aiming axes in early TFR trials in the Extrinsic group, but we did not observe it in the Intrinsic group (Extrinsic: $\boldsymbol{\mu} =—0.03\hat{\mathbf{i}} + 0.91\hat{\mathbf{j}} + 0.42\hat{\mathbf{k}}$, $\kappa = 49.25$, p <0.05, mean diff. = 18.8˚, Intrinsic: $\boldsymbol{\mu} = 0.06\hat{\mathbf{i}} + 0.77\hat{\mathbf{j}} + 0.63\hat{\mathbf{k}}$, $\kappa = 42.41$, p = 0.13, mean diff. = 4.1˚). The transfer of adaptation in the Extrinsic group is also evident from the change in velocity coordinates between late BL2 trials and early TFR trials (Fig 10a), as opposed to a similar velocity profile in the same stages of participants in the Intrinsic group (Fig 10b). This result indicates that, on average, the adaptation was represented in extrinsic, rather than intrinsic, coordinates. The small difference between late BL2 trials and early TFR trials in the Intrinsic group suggests that participants did not largely transfer the adaptation to an orthogonal initial hand orientation on average. However, the fact that this small difference is significant raises the possibility of learning in mixed coordinates. Nevertheless, this small effect vanishes after few trials (see Table A in S1 Appendix).

Participants in the Extrinsic group showed a clear, slow, and long-lasting transfer of adaptation (see Fig 9d (left), Table A in S1 Appendix). However, they did not transfer the full extent of adaptation, as indicated by the comparison of the aiming axes in early TFR trials with those

of late TRN trials rotated -90˚ around $\hat{\mathbf{i}}$ (Fig 9f (right), $\chi_2^2 = 50.59$, p $<$0.001, mean diff. = 26.05˚). This was somewhat unexpected. Indeed, we did not expect to see a complete 60˚ transfer, but we did expect to see the full extent of adaptation transferred. Moreover, this does not stem from the three-trials analysis, as a single-trial analysis resulted in similar statistics (see Table A in S1 Appendix).

Following the early TFR trials, the participants in the Extrinsic group gradually reversed their aiming axis until the late TFR trials (Fig 9e (left), $\chi_2^2 = 32.2$, p $<$0.001, mean. diff = 18.2˚). In fact, they regained BL2 behaviour, as indicated by the comparison of the aiming axes in late BL2 trials and late TFR trials (Fig 9f (left), $\chi_2^2 = 0.47$, p = 0.79, mean diff. = 2.4˚), and by the comparison of the angular velocity profiles in these stages (Fig 10a). Moreover, the mean of the aiming axes in the late TFR trials could indeed be the ideal BL2 axis ($\boldsymbol{\mu} = 0.71\hat{\mathbf{j}} + 0.7\hat{\mathbf{k}}$, $\kappa = 40.61$, p = 0.05, mean diff. = 0.76˚). Interestingly, it took participants more trials to regain BL2 behaviour during TFR than to adapt during TRN. As expected, participants in the Intrinsic group did not significantly change their aiming axis throughout the TFR session (Fig 9e (middle), Fig 10b, $\chi_2^2 = 1.51$, p = 0.52, mean diff. = 5.2˚), and regained a behaviour as in late BL2 trials (Fig 9f (middle), $\chi_2^2 = 6.99$, p $<$0.05, mean diff. = 8.3˚). By the end of the TFR session, the participants aimed near the ideal axis of BL2, as it did not largely differed from the mean of their aiming axes ($\boldsymbol{\mu} = 0.08\hat{\mathbf{i}} + 0.72\hat{\mathbf{j}} + 0.69\hat{\mathbf{k}}$, $\kappa = 52.25$, p = 0.22, mean diff. = 5.02˚).

**The perturbation was not generalized to an orthogonal target starting from the same initial orientation.** Our ability to determine whether the representation is consistent with one coordinate system rather than another relied on the fact that the generalization of the perturbation to other target directions was narrow enough. For that reason, the control group was set to determine the extent of generalization starting from a single initial orientation. We did not observe an aftereffect of adaptation to an orthogonal target (Fig 11a, bottom). Surprisingly, we observed a slight difference in the aiming axis between the late BL2 trials and the early TFR trials in the direction of the perturbation (Fig 9d (right), Fig 10c, $\chi_2^2 = 7.91$, p $<$0.05, mean diff. = 13.8˚). That said, the ideal BL2 axis did not largely differed from the mean of the aiming axes in the early TFR trials ($\boldsymbol{\mu} = -0.03\hat{\mathbf{i}} - 0.79\hat{\mathbf{j}} + 0.61\hat{\mathbf{k}}$, $\kappa = 47.29$, p = 0.09, mean diff. = 7.17˚). Additionally, no change was observed throughout the TFR session (Fig 9e (right), Fig 10c, $\chi_2^2 = 1.29$, p = 0.52, mean diff. = 3.9˚). This result characterizes the visuomotor rotation as having a narrow generalization to different target directions.

**The groups differed only in the transfer behaviour.** To check for differences between groups, we tested whether participants from different groups aimed differently in key trials. Since the targets in sessions of different groups were orthogonal, and in order to be able to compare the aiming axes, we rotated the aiming axes in the BL1 and TRN trials of the Extrinsic group and the aiming axes in the BL2 and TFR trials of the Control group by -90˚ around $\hat{\mathbf{i}}$. Participants in different groups responded differently to the initial exposure to the perturbation, possibly due to correction attempts performed by some of the participants in the Extrinsic group (Fig 9g (left), Extrinsic vs. Intrinsic: $\chi_4^2 = 5.89$, p $<$0.05, mean diff. = 14.6˚, Extrinsic vs. Control: $\chi_4^2 = 5.62$, p = 0.06, mean diff. = 8.5˚, Intrinsic vs. Control: $\chi_4^2 = 1.25$, p = 0.53, mean diff. = 6.3˚). Nevertheless, by the end of the TRN session, participants in all groups adapted a new aiming axis (Fig 9g (middle), Extrinsic vs. Intrinsic: $\chi_4^2 = 1.67$, p = 0.43, mean diff. = 5.6˚, Extrinsic vs. Control: $\chi_4^2 = 1.29$, p = 0.52, mean diff. = 5.1˚, Intrinsic vs. Control: $\chi_4^2 = 0.17$, p = 0.92, mean diff. = 2˚). In contrast, when the perturbation was removed and participants transferred to a new initial hand orientation, different groups utilized different aiming axes. This is due to the large difference between participants in the Extrinsic and both Intrinsic and Control groups (Fig 9g (right), Extrinsic vs. Intrinsic: $\chi_4^2 = 16$, p $<$0.001, mean diff. = 14.9˚,

Extrinsic vs. Control: $\chi_4^2 = 63.94$, p <0.001, mean diff. = 24.8˚, Intrinsic vs. Control: $\chi_4^2 = 12.18$, p <0.001, mean diff. = 11.2˚).

**The analysis of the aiming angles supports the extrinsic learning hypothesis, similar to the aiming axes analysis.** The time course of the aiming angles, averaged across participants, is depicted in Fig 11b. We performed a two-way mixed model ANOVA test to identify significant statistical differences in the aiming angles between participants from three different groups (Extrinsic, Intrinsic, and Control) and between key stages within the groups (late BL1, late BL2, early TRN, late TRN, early TFR, and late TFR). As in the spherical data analysis, we performed a sensitivity analysis on the number of trials that were used to define the aiming angle of each stage. The individual data of the three-trials analysis, as well as the mean aiming angles, are depicted in Fig 11c–11d. Here, we report the result of the three-trials analysis, and the full results of the sensitivity analysis is presented in S1 Appendix. The statistical analysis disclosed a significant main effect of the group, the trial and their interaction (Group: $F_{2,27} = 5.45$, $p < 0.05$, $\eta_p^2 = 0.29$, Trial: $F_{5,135} = 51.2$, $p < 0.001$, $\eta_p^2 = 0.65$, Group × trial: $F_{10,135} = 2.26$, p <0.05, $\eta_p^2 = 0.14$).

The multiple comparisons analysis revealed large, yet not statistically significant, changes in the aiming angle when the perturbation was initially applied, compared to the end of the BL1 session in all groups (Fig 11c, Extrinsic: $t_{27} = 3.18$, p = 0.05, d = 1.14, Intrinsic: $t_{27} = 2.16$, p = 0.59, d = 0.52, Control: $t_{27} = 2.76$, p = 0.15, d = 1.29). These large effects may be attributed to the averaging over three trials of a fast adaptation curve, and to correction attempts by participants due to continues visual feedback. As expected, the aiming angle increased throughout the TRN session in all groups (Fig 11c, Extrinsic: $t_{27} = 2.93$, p = 0.1, d = 1.15, Intrinsic: $t_{27} = 4.32$, p < 0.05, d = 1.02, Control: $t_{27} = 3.49$, p < 0.05, d = 1.48), yet a complete 60˚ change from the ideal BL1 axis was not achieved (Fig 11c, Extrinsic: 37.78˚, Intrinsic: 38.68˚, Control: 38.05˚). When transferred to an orthogonal initial hand orientation, participants in the Extrinsic group deviated in the direction opposite to the perturbation compared to the end of the BL2 session, as opposed to participants in the Intrinsic group (Fig 11c, Extrinsic: $t_{27} = 3.68$, p <0.05, d = 2.78, Intrinsic: $t_{27} = 0.73$, p = 1, d = 0.25). This shows that participants adapted to the perturbation by rotating around a new axis in extrinsic coordinates. In addition, we did not observe an aftereffect of adaptation in the direction opposite to the perturbation in the Control group, which indicates that the visuomotor rotation was not generalized to an orthogonally oriented target ($t_{27} = 2.71$, p = 0.18, d = 0.61). Surprisingly, a non-negligible, yet non-significant, deviation was observed to the other direction. By the end of the TFR session, participants in all groups behaved as in BL2 (Fig 11c, Extrinsic: $t_{27} = 0.13$, p = 1, d = 0.05, Intrinsic: $t_{27} = 0.28$, p = 1, d = 0.13, Control: $t_{27} = 1.94$, p = 0.94, d = 0.44). As such, the aiming angles in the Extrinsic group were reduced between the early and late TFR trials, while the aiming angles in the Intrinsic and Control groups did not largely change (Fig 11c, Extrinsic: $t_{27} = 5$, p <0.001, d = 1.47, Intrinsic: $t_{27} = 0.6$, p = 1, d = 0.17, Control: $t_{27} = 0.82$, p = 1, d = 0.36).

The groups differed only in the early TFR trials (Fig 11d, Extrinsic vs. Intrinsic: $t_{27} = 3.19$, p <0.05, d = 1.45, Extrinsic vs. Control: $t_{27} = 5.61$, p <0.001, d = 2.83, Intrinsic vs. Control: $t_{27} = 2.42$, p = 0.07, d = 0.97). The large difference between the Intrinsic group and the Control group is due to the negative deviation of participants in the Control group during the early TFR stage. All the other comparisons between groups in the three-trials analysis yielded non-significant and small differences (see S1 Appendix).

## Discussion

In our first experiment, we presented a novel approach for the analysis of the orientation path of a manipulated virtual rigid body, and used it to quantify the geodicity of rotations in an

orientation-matching task. By considering geodesics in $\mathbb{H}_1$, we showed that most of the participants tended to perform geodetic hand rotations. This means that they closely followed the great arc connecting the initial and final orientations of the hand. Nevertheless, a few participants performed non-geodetic orientation-matching movements, and the results of a few others were indecisive. The large variability within the participants and the small (but non-negligible) deviations from geodesics could be attributed to a few factors, including central planning. They could also be due to neuromuscular noise [38], as is evident from the dominance of stiffness in wrist rotation [37]. We also tested for the effect of task factors specifying the initial hand and target orientations. Our results indicated that manipulating these factors had only a small effect on the geodicity. Taken together, these results imply that the orientation of the hand is under kinematic control, yet the control is imperfect. The optimization of kinematics is also supported by our additional study that found a new power law that links the angular speed of a remote controlled tool that rotates around its center of mass with the local curvature of its orientation path [50]. Such optimization was observed in translational hand movements and is known as the speed-curvature power law [51], which was also observed in manipulation of remote controlled rigid bodies [52].

Our computational analysis design is based on the assumption that the nervous system represents the orientation of three-dimensional rigid bodies as quaternions. Quaternions are advantageous compared to other representations as their conversion from rotation matrices is nearly unambiguous. However, each rotation can be converted into two quaternions (**q** and −**q**). They also allow to define geodesics (see the Mathematical formulation section). However, quaternions may seem less intuitive compared to other representation, and are hard to visualize, as they are four-dimensional objects. Other representations of orientation include the helical axis representation which describes a moving rigid body as translating and rotating around an axis. Yet, helical axes are more sensitive to measurement errors than quaternions [53]. Euler angles are a parameterization of the rotational space using a series of rotations about three mutually orthogonal axes. Euler angles representation suffers from several disadvantages, among them is the ambiguity in conversion from a rotation matrix to a set of Euler angles, which impose challenges on defining metrics [54], and therefore geodesics. In addition, Euler angles suffer from Gimbal lock—the loss of one rotational degree of freedom when two rotations act about the same axis. The current experiment does not allow for differentiating between different representations of orientation in the nervous system. We chose the quaternions because they provide a convenient computational framework that facilitates further studies, as we showed in the second experiment.

Previous studies that focused on the control of orientation investigated similar rotational movements, but most of them applied joint constraints [37–43]. In our orientation-matching task, the participants were free to use all the degrees of freedom of their arm. In another study, the use of a mobile phone instead of a robotic device removed any joint constraint [44], yet it did not provide realistic three-dimensional visual feedback of the controlled phone. Instead, the orientation of the phone was projected onto the position of a point cursor. In contrast, we provided three-dimensional visual feedback. This made the task more natural, but also challenging to complete, as it required three-dimensional shape perception concurrent with object manipulation. This added complexity may have increased the variability of the orientation paths.

In our second experiment we demonstrated a newly developed paradigm, remapping the rotation of the hand using a visuomotor rotation. We tested whether or not participants are capable of compensating for such visual perturbation. We observed a fast, yet partial, adaptation in all the groups in the study. This was in contrast to what is known about adaptation to a visuomotor rotation with continuous visual feedback in point-to-point movements: the

adaptation is typically fast [55], but it is usually almost fully compensated [8, 24]. Early works found hints of separate mechanisms for the control of orientation compared to translation [31, 32, 35, 56, 57], and our results provide additional support for the idea of separate control of orientation and translation. However, we set the performance error of success at 15˚; thus it could be expected that participants would adapt to no more than 45˚, which was indeed the case (see the Result section).

Studies of planar translational movements have proposed that the adaptation process is composed of two components: implicit and explicit. An implicit process is an involuntary response to sensory prediction error [58] (i.e., the difference between the predicted and actual sensory outcome of a given motor command). An explicit process is an aware process in response to a performance error [20] (i.e., a signal that indicates task success or failure [59]) by deliberately re-aiming the hand such that it intentionally moves to a location that is distinct from the location of the target [60]. It was proposed that explicit processes are fast compared to implicit processes [61]. Adapting purely explicitly could explain the very fast, yet partial, adaptation that was observed in all the groups of our study. However, such an absence of an implicit process contradicts the establishment of an internal model used to counteract the perturbation, which is clearly evident from the transfer of adaptation observed in participants who trained and transferred to the same extrinsic target. Furthermore, the slow washout compared to the fast initial adaptation could not be explained purely by acting explicitly. Although implicit and explicit learning are separate processes [62], they may work concurrently or may be activated in a different phase of adaptation, which could explain the apparent contradiction between fast adaptation and slow washout that we observed in the current study. In future studies, the contribution of implicit and explicit learning may be tested using several methods, including clamped feedback [59], delayed feedback [63], limited reaction time [64] and aim reporting [62].

Our results showed that participants succeeded in compensating for the visuomotor rotation; this raised the question of whether they learned the perturbation in extrinsic or intrinsic coordinates. We observed a transfer of the adaptation when participants in the Extrinsic group transferred into a different initial hand orientation. Together with the absence of transfer in the Intrinsic group, it is implied that the learning occurred within an extrinsic coordinate system, rather than in an intrinsic one or in a combination of both. This result suggests that the visuomotor rotation alters the transformation remapping the hand's orientation in extrinsic coordinates onto its intrinsic coordinates, i.e., inverse kinematics, rather than the mapping between the intrinsic representation of the orientation and the muscle activation required to make successful rotations. Our result is in agreement with the extrinsic encoding of arm kinematics observed in translation movements [4, 5, 20, 24]. However, the variability in the aiming axes in the first TFR trial of the Intrinsic group may indicate a mixed coordinate model, as previously suggested in reaching movements [26]. This is less likely given our data since the effect vanished fast, as seen in the three-trials TFR analysis.

To test in which coordinate system the visuomotor rotation was represented, we chose the ideal rotation axes to be orthogonal between the Extrinsic group and the Intrinsic group during training. As a result, participants in these two groups rotated towards different targets. This raises the question of whether the trained target had an effect when transferring to a new initial orientation. That is, it may be that the transfer of adaptation that we observed in the Extrinsic group stems from the specific training target, regardless of the trained coordinates. Nevertheless, our study first and foremost established a new viosuomotor rotation methodology. Furthermore, our results need to be further validated with many more rotation axes, initial hand orientations, and targets to ascertain the robustness of our finding [25].

Both the spherical aiming axes analysis and the scalar aiming angles analysis suggested that the adaptation of the rotation-based visuomotor rotation took place within extrinsic coordinates. This is suggested from the results of the three-trials analysis, in which participants in the Extrinsic group showed a large aftereffect in the transfer of the adaptation to a new initial orientation, while participants in the Intrinsic group and the Control group did not. Along with the large difference between the Extrinsic group and the Intrinsic group in the early TFR stage, this result supports the extrinsic learning hypothesis (see Fig 4). We chose to report the results of the three-trials analysis as it nicely captured both the learning and transfer effects, and is less prone to noise than a single-trial analysis. However, while this choice is compatible with the slow washout observed in the Extrinsic group, it may underestimate the fast learning in the early TRN stage of all groups. Nevertheless, even the single-trial analysis was able to capture a large difference between the Extrinsic group and the Intrinsic group in the early TFR stage, though this difference was not statistically significant. The choice of the number of trials to average is somewhat arbitrary, therefore, we provided the full results of the sensitivity analysis for both the spherical aiming axes analysis and the scalar aiming angles analysis (see S1 Appendix). Furthermore, we report differences within and between groups using effect sizes, rather than just p-values, since a significant difference is not necessarily a large difference.

While our results suggest that humans represent the orientation of rigid bodies that are manipulated by the hand in extrinsic coordinates, previous studies have suggested that cognitive perception may change these coordinates [18]. One way to test this hypothesis is to associate the controlled cursor with either one of the reference frames during adaptation and test for the effect on the transfer of the adaptation. This could possibly be done by attaching a three-axis frame to the cursor, or by showing the entire arm in the experimental scene rather than just the manipulated object.

In this study we employed a visuomotor rotation for studying adaptation in object manipulation. Classical studies of translational movement employed other types of perturbations, including adaptation to dynamical perturbations, such as viscous force fields [7]. Quaternion-based metrics can also be used in the analysis of rotational hand movements in the presence of dynamical perturbations, such as torque fields. Under such interference, the hand is expected to largely deviate from the geodetic quaternion path. One way to measure such deviation would be to integrate over the quaternion path by defining a curve between each quaternion sample and the geodesic, which intersects with the latter orthogonally. Then, using an interpolation method similar to trapezoidal integration, the integral may be computed. To account for the variability in the trial's duration, the integral may also be time-normalized.

Beyond pure theoretical interest, movement representations have important practical implications in many fields. One such field is teleoperation. During teleoperation, users manipulate a leading robotic manipulator to control the movement of a follower manipulator. The follower is usually viewed through one or more cameras. Controlling the follower primarily requires that the user plan a desired movement in the follower's reference frame. Then, to determine the required motion of the leader device, the user must be able to transform that frame into the leader's reference frame. This mental transformation becomes difficult when the frames are misaligned [39], potentially affecting the performance of the user [65]. Our results contribute to a deeper understanding of how humans may represent and compensate for visual remapping in teleoperation systems. Moreover, they highlight the importance of appropriate planning of the rotational workspace [66]. The visuomotor rotation discussed in this paper is relevant for robotic teleoperation if the leader-follower frames or camera viewpoint are misaligned due to failures of the operator to control the camera viewpoint, misreading by sensors, or an incorrect kinematic model of the robot [57, 67–70]. The previously reported difficulty in compensating for such distortions [57] is consistent with the partial

adaptation that we observed in our visuomotor rotation experiment. Moreover, the geodicity score that we developed may complement known rotation-based metrics that quantify human motor skills during teleoperation [71].

Additionally, our novel rotation-based visual perturbation has implications in the field of neurorehabilitation. Robotic neurorehabilitation utilizes motor tasks to induce neural plasticity that will improve motor function and help in recovery from post injury loss of motor abilities [72, 73]. For example, a recent study demonstrated the ability to personalize robotic rehabilitation training and monitor motor improvement in reaching by linking it to visuomotor adaptation [74]. However, there have also been many reports concerning the lack of generalization from such oversimplified movements to activities of daily living [75]. Perturbation of rotational variables could be another approach that would lead to motor improvement in patients with wrist rotation dysfunction, and could be included in recovery treatment schedules [38, 39]. However, previous attempts projected wrist rotations into movements of a planar cursor, and it is possible that manipulation of realistic objects with three degrees of freedom, as implemented in the current study, could contribute to better generalization to real-life activities.

## Conclusions

The methodology presented here provides a novel three-dimensional approach to study how humans control the orientation of rigid bodies. The experiments reported here utilized this methodology to establish a basic understanding regarding the control of the orientation of rigid bodies. We characterized orientation-matching movements by their geodicity in $\mathbb{H}_1$, and found evidence of kinematic control. We also studied adaptation to a visuomotor rotation applied on the rotation of the hand, rather than on its translation. We found that the sensorimotor system adapts to this visuomotor perturbation, probably by forming an extrinsic representation of the new mapping between the orientation of the hand and the orientation of the controlled object, and uses it to compensate for the visuomotor rotation. Understanding the control of the orientation of objects in three dimensions is an important and insufficiently studied aspect of the control of movement, with many practical applications such as the control of human-centered teleoperation systems and neurorehabilitation.

## Supporting information

**S1 Appendix. Full results of the sensitivity analysis on the aiming axes and angles.** Tables A—C present the results of the sensitivity analysis that we performed on the number of averaged trials in the aiming axes analysis of experiment 2 (Table A—comparison within groups, Table B—comparisons between groups, and Table C—tests for the mean axis). Tables D—F present the results of the sensitivity analysis that we performed on the number of averaged trials in the aiming angles analysis of experiment 2 (Table D—ANOVA results, Table E—comparisons within groups, and Table F—comparisons between groups). $^*p < 0.05$, $^{**}p < 0.001$. (PDF)

## Author Contributions

**Conceptualization:** Or Zruya, Ilana Nisky.

**Data curation:** Or Zruya.

**Formal analysis:** Or Zruya.

**Funding acquisition:** Ilana Nisky.

**Investigation:** Or Zruya, Ilana Nisky.

**Methodology:** Or Zruya.

**Project administration:** Or Zruya.

**Resources:** Ilana Nisky.

**Software:** Or Zruya.

**Supervision:** Ilana Nisky.

**Validation:** Or Zruya, Ilana Nisky.

**Visualization:** Or Zruya.

**Writing – original draft:** Or Zruya, Ilana Nisky.

**Writing – review & editing:** Or Zruya, Ilana Nisky.

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
