## [Decision Letter · Decision Letter 0]

13 Jul 2022

Dear Zruya,

Thank you very much for submitting your manuscript "Orientation control strategies and adaptation to a visuomotor perturbation in rotational hand movements" for consideration at PLOS Computational Biology.

As with all papers reviewed by the journal, your manuscript was reviewed by members of the editorial board and by several independent reviewers. The Reviewers appreciated the novel computational approach, but had a number of concerns and suggestions which I believe will help to improve the paper. In light of the reviews (below this email), we would like to invite the resubmission of a significantly-revised version that takes into account the reviewers' comments.

We cannot make any decision about publication until we have seen the revised manuscript and your response to the reviewers' comments. Your revised manuscript is also likely to be sent to reviewers for further evaluation.

Sincerely,

Adrian M Haith

Associate Editor

PLOS Computational Biology

Wolfgang Einhäuser

Deputy Editor

PLOS Computational Biology

Reviewer's Responses to Questions

**Comments to the Authors:**

Reviewer #1: General

Here the authors use a new experimental paradigm to examine how the sensorimotor system adapts to visual rotations that have been perturbed about a 3D axis. The authors use quaternions to perform the kinematic analyses, which have benefits over more common and traditionally used affine transformation approaches (Euler, Cardan, helical axis). The authors cleverly exploit that quaternion multiplication is not commutative to get at the notion that humans adapt in extrinsic verses intrinsic coordinate frames. Overall, I am generally positive as there is merit to their new approach, which is largely a methods paper and differs from the commonly used visuomotor rotation paradigm. Comments, questions, and concerns are mentioned below. Can the authors please respond in a point-by-point manner.

1. For the extrinsic and intrinsic groups in Experiment 2, the two groups rotated about two very different axes and performed very different movement patterns during training. A concern here is that a history of being reinforced along very different axes of rotation and movement patterns may have had different carry-over effects into the transfer task, irrespective of learning an intrinsic or extrinsic coordinate system. That is, given how different the rotations / movements were during training, this would seem like a potential confound when comparing groups in the transfer task. Further, in both groups participants must adapt the intrinsic coordinates to carry out the task, since the upper arm was left completely unconstrained. Since this is a new technique and the authors’ suggested main finding hinges on two conditions with very different movement patterns during training, it would have been useful to test many more axis rotations to be assured that the results are not merely a coincidence but a robust finding. Can the authors please comment and include any changes to the text that might bolster their suggestion.

2. The authors use only one data point (e.g., trial 301) for several of the comparisons, which is likely quite suspectable to noise. The adaptation does occur quite quickly, so perhaps taking the median of three trails for all the points of interest (early learning, late learning, early transfer, etc.) to create more robust statistics. This is highlighted when the authors do not have statistical significance between the extrinsic verse intrinsic groups (lines 651-656) with a single trial analysis, which is critical to their suggestion that humans use an extrinsic coordinate system (global) as opposed to an intrinsic coordinate system (e.g., joint or muscle space) to adapt to the perturbed 3D rotations. It is not sufficient that there are within participant differences between training and transfer, as currently shown in the Figure 8, to make this suggestion. That said, significance (p < 0.05) is an arbitrary line in the sand. Further, the authors do find significance when considering three trials.

a. Instead, the authors should focus on reporting effect sizes (also reported on the figures).

b. Perform a sensitivity analysis using different numbers of trials (e.g., 1-10 trials)

c. They should also include a figure (e.g., 8D) that shows the group comparisons (intrinsic vs extrinsic vs control) for the transfer portion.

d. Further, they should include text in the discussion that highlights this limitation.

e. In this case, it would be useful to including the supplementary in the main text since the two metrics support one another.

3. The manuscript would benefit from plotting more dependent measures. The authors do report overall adaptation angle on a trial-by-trial level. It would be useful to plot the angles/velocity over time for each group, during key trials (e.g., early adaptation). Perhaps this is difficult with the instantaneous axis, but one could conceivably use the average axis. The authors do report individual angular velocities along each DoF, but it would be useful to also plot their Euclidean norm. Another useful metric here would be movement times.

4. The authors should highlight the advantages of quaternions over traditional techniques (no gimbal lock / singularities and less impacted to noise), which the authors should highlight. The obvious downside of quaternions is that they are much harder to interpret and visualize compared to other common techniques, such as Euler, Cardan, Helical axis. Of these, quaternions are like helical axis in spirit, which are much easier to visualize and have been used on the biomechanics literature in the past (Woltring, 1984). Perhaps it is worthwhile to briefly discuss the advantages and disadvantages to their approach.

5. A strength of the paper is that the authors have made a new paradigm to study visuomotor rotations. As such, they have derived a nice metric to define the path traversed by the quaternion vector relative to the geodesic. The authors define the geodesics metric, which is quite nice. Perhaps another useful metric to construct is the integral of the deviation from the geodesic, normalized by time.

6. Was the time derivative taken on the I,j,k, components to calculate angular velocity?

7. abstract: “We found that rotations are generally performed by following a geodesic in the quaternion hypersphere, which suggests that, similarly to translation, the orientation of the hand is centrally controlled.” It would be useful to have a follow-up sentence here, making the link to why the shortest path indicates central control.

8. lines: 10-14: there are several non-kinematic hypotheses that explain this ‘invariant’ (which does alter shape depending on the task). For example, min torque rate (Uno), OFC, where it arises by minimizing effort costs in the presence of signal dependent noise, as well as energetics (Wong, Kuo, 2021). Perhaps change the language here slightly to be more encompassing.

9. Useful to include the following citation, which examine sensorimotor control using quaternions: Leclercq, G., Lefèvre, P., & Blohm, G. (2013). 3D kinematics using dual quaternions: theory and applications in neuroscience. Frontiers in behavioral neuroscience, 7, 7.

10. Do the instantaneous axes always intersect at the same point?

11. line 335: Include baseline trials. It is always useful to see participants initial behaviour.

12. It will be very important to provide the code, since using quaternions is quite involved.

Sincerely,

Joshua Cashaback

Reviewer #2: Many previous studies have focused on the planar reaching movement to investigate problems such as how the motor system plans the trajectory or adapts the movement to a novel environment. The present study has challenged examining hand orientation control, which is important yet not extensively studied. To this end, the authors developed a novel robotic device and VE for the hand orientation-matching task and adopted quaternion-based analysis to evaluate the kinematics.

Exp.1 demonstrated that the QGS value was close to the optimal value in most cases, suggesting that the hand orientation was controlled efficiently like a straight movement path observed in the reaching movement. The authors also tried to impose visual rotation perturbation to this task and based on the generalization pattern after the adaptation, they concluded that the adaptation was likely to be achieved in the extrinsic coordinates.

This is a very interesting study tackling the problem of hand orientation control by using a novel experimental paradigm. This methods will enable to investigate a lot of interesting research

I also felt that adopting quaternion-based data analysis and the task design (initial and target orientation etc.) was very smart. However, I was concerned with several points, particularly about the readability of the paper. Below, I would like to raise several issues that the authors need to consider.

Major points

1. It was too difficult to understand the task settings. There are several unclear or confusing points. My interpretation of Experiment 1 is as follows, but I am unsure if it is correct.

There are two initial orientations and two target orientations (Table 1). When the initial orientation was given (o1 or o2 condition), the target orientation was determined by rotating the initial orientation around the axes (e.g., \\hat{i} or -\\hat{j}, etc. in Table 1) by \\alpha (40~60 deg). There are also two ways of rotation (intrinsic and extrinsic described in Eqs. (3) and (4)). In contrast, when the target orientation was given (t1 or t2 condition), the initial orientation was determined by inversely rotating the target orientation.

Anyway, the task settings should be described more clearly. Otherwise, I believe that almost all readers should be confused about the task.

2. Expression using the quaternion is intelligent and impressive, but I am not sure how this expression is common in the motor learning research field. Thinking about the movement pattern from quaternions might be very hard. Displaying visual examples of the initial and target orientations for several conditions would help understand the difference between aligned and misaligned axes and between intrinsic and extrinsic rotations.

3. Did not the author observe the learning effect in Experiment 1? Performing the task in different condition seems not easy as represented by the larger QGS (Fig.6).

4. This might be related to the point #3 described above. In Exp.1, there were no statistical differences in the QGS values among various conditions. However, was it possible that the absence of significant difference resulted from the interference between intrinsic and extrinsic rotations? I was interested in how the large QGS trial took place. It seems that the large QGC would be generated by the transition from intrinsic to extrinsic rotation and vice versa?

5. I wondered how stable the baseline trials in Exp.2 were. The QGS values were maintained low? If the movement in the baseline was not stable enough (as shown in Fig.6), the data of adaptation experiment was not reliable.

Minor points

Line 383 “Early Transfer”: Figure 4 used “First Transfer”.

Line 555 “Fig.8a”: Fig.8a does not show any data of the last BL1 and that of the first TRN. Fig.8a should be Fig.8c?

Lines 560-562: The sentence seems inconsistent with the data. Fig.8c shows no significant difference in the aiming direction between the last BL1 and the first TRN.

Line 573 “Fig.8a”: Again, Fig.8 does not show any data of the first TRN and that of the last TRN. Fig.8a should be Fig.8c?

Line 604 “Fig.8a”: Fig.8a should be Fig.8c

**Have the authors made all data and (if applicable) computational code underlying the findings in their manuscript fully available?**

Reviewer #1: Yes

Reviewer #2: Yes

PLOS authors have the option to publish the peer review history of their article (what does this mean?). If published, this will include your full peer review and any attached files.

Reviewer #1: No

Reviewer #2: No
---

## [Decision Letter · Decision Letter 1]

14 Nov 2022

Dear Zruya,

We are pleased to inform you that your manuscript 'Orientation control strategies and adaptation to a visuomotor perturbation in rotational hand movements' has been provisionally accepted for publication in PLOS Computational Biology.

Best regards,

Adrian M Haith

Academic Editor

PLOS Computational Biology

Wolfgang Einhäuser

Section Editor

PLOS Computational Biology

Reviewer's Responses to Questions

**Comments to the Authors:**

Reviewer #1: The authors have done a very thorough job revising the manuscript, addressing my comments, and highlighting limitations within the manuscript. I just have a few minor comments remaining.

1. For the figures (e.g., Fig 11), it would be useful to display both p-values and effect size together, since most readers will be interested in both.

2. Perhaps I missed this, but why not report a cohen’s d instead of mean of the angular distance? Mean difference doesn’t account for the standard deviation like effect size.

3. Regarding the following additional paragraph, "In this study we employed....the integral may also be time-normalized" I was convinced by the argument to not include the integral metric, so this portion of text is not necessary for the manuscript.

Sincerely,

Joshua Cashaback

Reviewer #2: I am pleased the authors have seriously considered and addressed my suggestions and questions. I believe the revised paper improves readability considerably. I think using quaternions to represent wrist movements is a clever idea. I have no further comments or suggestions.

**Have the authors made all data and (if applicable) computational code underlying the findings in their manuscript fully available?**

Reviewer #1: Yes

Reviewer #2: Yes

PLOS authors have the option to publish the peer review history of their article (what does this mean?). If published, this will include your full peer review and any attached files.

Reviewer #1: **Yes: **Joshua Cashaback

Reviewer #2: No

---

## [Editor Report · Acceptance letter]

24 Nov 2022

PCOMPBIOL-D-22-00811R1 

Orientation control strategies and adaptation to a visuomotor perturbation in rotational hand movements

Dear Dr Zruya,

I am pleased to inform you that your manuscript has been formally accepted for publication in PLOS Computational Biology. Your manuscript is now with our production department and you will be notified of the publication date in due course.

With kind regards,

Zsofi Zombor
